# MM-OPERA: Benchmarking Open-ended Association Reasoning for Large Vision-Language Models

**Zimeng Huang**[1,2], **Jinxin Ke**[1], **Xiaoxuan Fan**[3], **Yufeng Yang**[1], **Yang Liu**[1], **Liu Zhonghan**[1],
**Zedi Wang**[1], **Junteng Dai**[1], **Haoyi Jiang**[1], **Yuyu Zhou**[3], **Keze Wang**[1], **Ziliang Chen**[2*]

[1]Sun Yat-sen University
[2]Peng Cheng Laboratory
[3]Jinan University

{huangzm29, kejx, liuzhh268, wangzd6, daijt3, jianghy55}@mail2.sysu.edu.cn,
{yangyf226, liuy856, wangkz}@mail.sysu.edu.cn, fanxx@stu2022.jnu.edu.cn,
zyy@jnu.edu.cn, c.ziliang@yahoo.com

## Abstract

Large Vision-Language Models (LVLMs) have exhibited remarkable progress. However, deficiencies remain compared to human intelligence, such as hallucination and shallow pattern matching. In this work, we aim to evaluate a fundamental yet underexplored intelligence: association, a cornerstone of human cognition for creative thinking and knowledge integration. Current benchmarks, often limited to closed-ended tasks, fail to capture the complexity of **open-ended association reasoning** vital for real-world applications. To address this, we present MM-OPERA, a systematic benchmark with 11,497 instances across two open-ended tasks: Remote-Item Association (RIA) and In-Context Association (ICA), aligning association intelligence evaluation with human psychometric principles. It challenges LVLMs to resemble the spirit of divergent thinking and convergent associative reasoning through free-form responses and explicit reasoning paths. We deploy tailored LLM-as-a-Judge strategies to evaluate open-ended outputs, applying process-reward-informed judgment to dissect reasoning with precision. Extensive empirical studies on state-of-the-art LVLMs, including sensitivity analysis of task instances, validity analysis of LLM-as-a-Judge strategies, and diversity analysis across abilities, domains, languages, cultures, etc., provide a comprehensive and nuanced understanding of the limitations of current LVLMs in associative reasoning, paving the way for more human-like and general-purpose AI. The dataset and code are available at https://github.com/MM-OPERA-Bench/MM-OPERA.

## 1 Introduction

Recent advancements in Large Vision-Language Models (LVLMs) have significantly improved their ability to handle multi-modal inputs and address diverse tasks. Systems such as GPT-4 [69], Gemini models [79], and LLaVA [55] exhibit remarkable proficiency in visual understanding, language generation, and multi-step reasoning. These capabilities are driving transformative applications across fields such as education, design, scientific discovery, embodied intelligence, and so on [39, 19, 30, 57].

Existing benchmarks for LVLMs [3, 62, 35, 95, 44, 43, 96, 18, 31, 59, 60] has facilitated systematic assessments of instruction-following and alignment tasks, focusing on recognition, comprehension, and reasoning. However, the evaluation of association intelligence in LVLMs remains underexplored.

---

*Correspondence to: Ziliang Chen

39th Conference on Neural Information Processing Systems (NeurIPS 2025) Track on Datasets and Benchmarks.

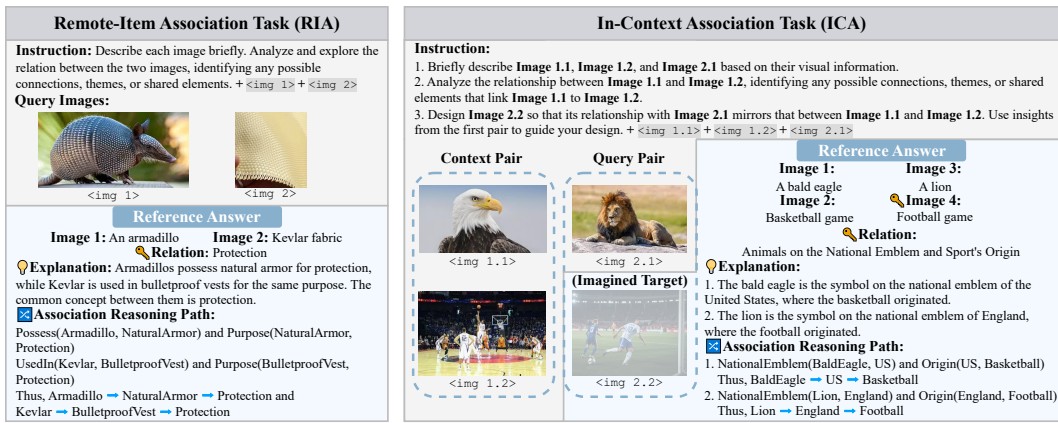

Figure 1: An overview of MM-OPERA. The RIA task challenges models to discover meaningful connections between unrelated elements, while the ICA task requires transferring relationship patterns from a context pair to a query item to generate an appropriate target. The reference answer represents just one possible valid response. The association reasoning paths are used to evaluate the coherence and depth of the step-by-step reasoning process.

**Association**, a cornerstone of human cognition, enables creative thinking [64], underpins the integration of fragmented information into coherent knowledge and supports critical cognitive processes such as memory, perception, and rule discovery [5]. We argue that LVLMs need to develop this core capability to move beyond shallow pattern matching toward true knowledge synthesis and reasoning. It is a prerequisite for many real-world applications such as scientific discovery, creative ideation and design, personalized education, innovative problem-solving and robot planning.

Current efforts, such as the Labyrinth of Links [46] have begun to formalize association as an evaluation target, using closed-ended tasks with predefined options to probe associative memory. While this approach offers valuable insights, it falls short of capturing the full scope of association reasoning required for real-world AI applications. We argue that *open-ended association reasoning* is essential for two key reasons: (1) Closed-ended tasks with fixed options may introduce bias, subtly guiding the model's associative behavior and masking its true capacity for independent reasoning; (2) The fixed-answer format struggles to evaluate complex, long-form association reasoning, limiting the ability to challenge models on intricate, multi-step relational inference. These limitations motivate our development of a new benchmark that prioritizes **open-endedness** to rigorously assess and ultimately enhance LVLMs' association reasoning capabilities.

In cognitive science, association emerges from the interplay of *convergent and divergent thinking*: the former identifying meaningful connections and selecting optimal solutions; the latter generating multiple unique ideas [83, 13, 63]. The Remote Associates Test (RAT) [64, 24, 17, 87, 2] exemplifies this by requiring individuals to uncover links between distant concepts, a process vital for adaptive problem-solving. To mirror this in LVLMs and address the shortcomings of prior work, we propose **MM-OPERA** (**M**ulti-**M**odal **OP**en-**E**nded **R**easoning-guided **A**ssociation), a benchmark designed to evaluate association reasoning without predefined constraints. It assesses how models identify and express meaningful links across distant concepts (*i.e.* convergent thinking), expected to emerge through diverse reasoning paths (*i.e.* divergent thinking). Table 1 highlights how MM-OPERA diverges from The Labyrinth of Links by adopting open-ended tasks, more challenging reasoning scenarios, and a broader scope of evaluation, enabling a deeper probe into LVLMs' relational inference abilities.

MM-OPERA comprises 11,497 instances across two core tasks (Figure 1): *Remote-Item Association* (RIA), testing the ability to link distant concepts with structured reasoning, and *In-Context Association* (ICA), probing pattern recognition within in-context learning [29]. Spanning 13 associative dimensions and diverse cultural, linguistic, and thematic contexts, it offers a comprehensive evaluation framework. It prioritizes free-form responses, employing reference answers as heuristic quality benchmarks rather than rigid correctness criteria. To evaluate open-ended outputs, we design tailored LLM-as-a-Judge strategies with a cascading scoring rubric. Furthermore, by leveraging process-

Table 1: Comparison between The Labyrinth of Links and MM-OPERA.

| Dimension | The Labyrinth of Links | MM-OPERA (Ours) |
|---|---|---|
| **Task Format** | Multi-choice, closed-ended | **Free-form, Open-ended** |
| **Association Tasks** | Basic Steps: Single / Synchronous / Asynchronous | **More Complex:** Remote-Item Association / In-Context Association |
| **Association Scope** | Adjectives and Verb *limited semantic concepts* | **3 relationship types, 13 ability dimensions;** *broad cultural, linguistic and thematic contexts* |
| **Evaluation Metrics** | Correctness-focused: Max / Mean Step, Success Ratio | **Multi-dimensional assessment:** Score Rate, High Score Rate, △HR, Reasoning Score, Reasonableness, Distinctiveness, Knowledgeability |
| **Evaluation Flexibility** | Option-based, limited generative capacity | **Fully generative,** *supports diverse reasoning paths and rationales* |

reward principles to trace reasoning steps, our evaluation captures cognitive flow and knowledge integration, surpassing traditional outcome-focused metrics.

Our contributions are threefold:

1. **MM-OPERA:** We introduce a benchmark of 10,000+ instances for evaluating LVLMs' association reasoning, centered on Remote-Item Association (RIA) and In-Context Association (ICA) tasks inspired by classic psychometric studies. It spans 13 analytical dimensions to enable comprehensive assessment.

2. **LLM-as-a-Judge Strategies:** To support open-ended evaluation, we design tailored LLM-as-a-Judge methods that assess both response quality and reasoning processes, enabling fine-grained and reliable scoring.

3. **Profound Findings:** Our analysis reveals key limitations of current LVLMs and highlights the critical role of association reasoning in advancing real-world, general-purpose AI.

## 2 Related Work

**Large Vision Language Models (LVLMs).** Early studies [78, 98, 73] established the foundations of vision-language models. CoCa [94], Flamingo [1], and BLIP-2 [47], advanced performance with enhanced architectures and large-scale multimodal pretraining. InstructBLIP [25], MiniGPT-4 [103], and LLaVA [56], have refined multimodal instruction tuning and alignment strategies. Recent open-source LVLMs, *e.g.*, LLaVA-OneVision [45], mPLUG-Owl3 [91], and Qwen2-VL [7], have extended these capabilities to multi-image and video understanding. Proprietary models like GPT-4V [69], Gemini-Pro-V [79], and Qwen-VL-Max [8] have demonstrated state-of-the-art performance.

**LVLM Benchmarks.** The evaluation of LVLM has progressed from early benchmarks like VQA [3, 35] and OK-VQA [62] to broader assessments such as SEED-Bench [44], LAMM [92], LVLM-eHub [90], MMBench [59], MSCOCO [53], and MM-Vet [95], covering tasks like Optical Character Recognition (OCR) [58], adversarial robustness [100], and hallucination detection [23, 54, 48, 84]. Specialized benchmarks target various capabilities: MathVista [61], CLEVR [40], CVR [97], ReMI [42], Encyclopedic VQA [65], LogicVista [89], SPACE [74], BLINK [32], ZeroBench [75], MMMU [96], and Visual Riddles [15] each focus on different aspects of reasoning and perception. Li et al. [46] propose an adjective-verb association benchmark, but it is constrained to predefined categories, leaving open-ended associative reasoning largely unexplored.

**Psychometric Test for AI Evaluation.** Researchers have proposed psychometric frameworks to assess AI cognition [38], ranging from personality and theory-of-mind benchmarks [49], latent trait profiling [72], and reasoning evaluation via the Technology Acceptance Model (TAM) [50], to broader construct-oriented approaches emphasizing underlying cognitive mechanisms over task-level performance [86]. Adaptive testing further enhances efficiency by dynamically adjusting to model responses [104]. Association reasoning has also been modeled and involved in AI evaluation [76, 46]. Mednick's Theory of Creativity defines creativity as forming remote connections [64], underpinning associative creativity theories [71, 10] and the Remote Associates Test (RAT), adapted for semantic and visual associations [16, 68, 11, 67] through convergent thinking tasks. Divergent thinking [81, 14] is also assessed via tasks like the Alternate Uses Task (AUT) [36] and Divergent Association Task

(DAT) [66]. Studies on LLMs and LVLMs reveal mixed results: "leap-of-thought" tasks enhance divergent reasoning [102], GPT models show varied creativity and even surpass humans [41, 22].

# 3 MM-OPERA: Dataset

In this section, we illustrate the task design and the corresponding dataset of MM-OPERA. Section 3.1 elaborates association tasks and Section 3.2 presents the dataset statistics. The data curation is detailed in Appendix A.4.

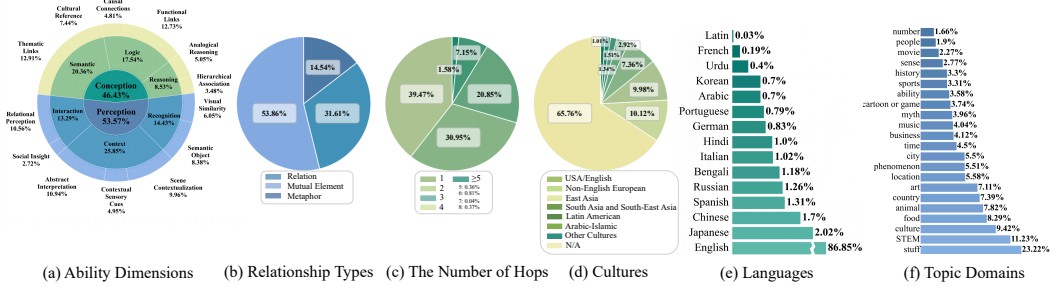

Figure 2: Statistics of MM-OPERA. (a) Hierarchical ability taxonomy consists of 3 levels, refining perceptual and conceptual associations. We report each ability's frequency as a percentage of total label occurrences to better represent the dataset's distribution. (b) Three relationship types capturing diverse associative connections. (c) The number of hops in the association reasoning path, quantifying different associative reasoning complexity. (d) Different cultures, (e) 15 languages, and (f) 22 topic domains ensuring broad cultural, linguistic, and thematic diversity.

## 3.1 Association Tasks: Motivation and Definition

Associative ability is commonly assessed through the Remote Associates Test (RAT), which presents participants with three seemingly unrelated items and asks them to identify a fourth item that connects all three. While RAT offers psychological validity, it primarily emphasizes instinctive convergent thinking with a single-hop reasoning path across items. However, the human response time metric in RAT is difficult to replicate in machines. Moreover, RAT lacks the complexity needed to capture the divergent thinking process that underlies convergent thinking.

To remedy this, we re-develop the remote-item paradigm into two novel association tasks, both incorporating a chain of thought with multi-step reasoning structure across a pair of remote multimodal items. LVLM is required to generate an open-ended answer with the explanation, while the reference answer and its underlying reasoning chain is provided for newly invented LLM-as-a-Judge strategies (Section 4).

**Remote-Item Association.** The RIA task instance challenges LVLMs to discover meaningful links between seemingly unrelated elements across text, images, or mixed modalities. As shown in Figure 1 (left), when presented with query images of an armadillo and Kevlar fabric, an LVLM candidate is demanded to identify their shared protective function—moving beyond surface features to reveal conceptual bridges. This task encourages cross-domain reasoning and rewards both logical coherence and creative insight, as multiple valid associative explanations may exist.

**In-Context Association.** The ICA task instance extends RIA to in-context learning, thus evaluating a model's ability to recognize, abstract, and extend associative patterns within a creative framework. In Figure 1 (right), the model first identifies the connection between a bald eagle and basketball (America's national symbol and a sport originating there), then applies this pattern to generate the appropriate complement to a lion image (football, as England's national symbol relates to the sport's origin). This task tests the model's pattern-based reasoning, ability to abstract cross-domain associations and balance creative flexibility with logical consistency.

## 3.2 Dataset Statistics

MM-OPERA contains 11,497 task instances (8,021 in RIA and 3,476 in ICA) spanning diverse modalities, concepts, and reasoning complexities. Its comprehensive design supports thorough evaluation of LVLMs' associative capabilities across multiple dimensions, reflecting the multifaceted nature of human associative reasoning. Detailed statistics are presented in Figure 2.

**Sample Distribution and Design.** The RIA dataset includes Multiple-Image variants where identical concepts appear in different images, enabling controlled sensitivity testing of LVLMs' visual perception. Notably, over 25% of these instances exhibit unique concept pairs, ensuring breadth in conceptual coverage. The ICA dataset employs an circular evaluation strategy, where each set of four images generates four distinct questions, each requiring the model to reason about one image based on the relationships established by the other three.

**Hierarchical Ability Taxonomy.** Associative thinking operates on multiple cognitive levels, from perception to conception—both crucial for understanding complex environments [88, 77, 37, 12]. Perception handles immediate sensory inputs, while conception deals with abstract, knowledge-driven associations. These fundamental processes form our Level-1 (*L-1*) associative ability. We further refine it into six *L-2* and thirteen *L-3* dimensions, creating a hierarchical framework that mirrors human cognition and enables systematic evaluation of LVLMs' capabilities in processing both sensory input and abstract reasoning. Detailed definitions are in Appendix A.2.

**Types of Relationship.** To capture the nuanced ways entities or concepts connect, we identify three relationship categories: *Relation*, denoting general links between entities; *Mutual Element*, indicating shared traits; and *Metaphor*, connecting entities through abstract or symbolic meanings. This tripartite classification enhances the benchmark's ability to evaluate associative reasoning across both literal and abstract dimensions, reflecting the multifaceted nature of human associative thinking.

**Association Reasoning Path.** While natural language explanations offer valuable insights into associative reasoning, they often lack the structured clarity needed to systematically evaluate complex reasoning processes. To address this limitation, we introduce *Association Reasoning Paths*, a visual framework that represents the reasoning process as a directed path with arrows connecting concepts. Each *hop* in this path represents a discrete reasoning step, with the total number of hops directly reflecting the association's complexity. For instance, connecting an armadillo to Kevlar might require a four-hop path through intermediate concepts leading to their shared protective function (Figure 1). This structured representation enables reasoning-guided evaluation detailed in Section 4.2.

**Diversity.** Our dataset deliberately incorporates various cultures, 15 languages with their unique linguistic devices (idioms, puns, proverbs) as association links, and 22 topic domains. This diversity is proposeful: while LVLMs possess vast knowledge repositories from their training, true intelligence lies in the ability to activate these knowledge pathways—connecting observations to prior knowledge across cultural, linguistic, and domain boundaries—which is the basis of association.

## 4 LLM-as-a-Judge Strategies for MM-OPERA

While open-ended tasks eliminate potential hints that might influence models' association behaviors, they present significant evaluation challenges. Traditional methods including human evaluation, rule-based systems, and automatic metrics, often struggle with inconsistency and bias when assessing such unconstrained responses [4, 21]. To address these challenges, we present three complementary LLM-as-a-Judge strategies: Section 4.1 introduces our Regular Scoring framework, which serves as the foundation for Process-Reward Evaluation in Section 4.2. Evaluation prompts are available in Appendix E.

### 4.1 Regular LLM-as-a-Judge Scoring

Since the open-ended responses and references answers presented in text, we adopt LLMs as automatic judge engine. Unlike prior benchmarks that use per-sample criteria [33], we adopt unified scoring rubrics that evaluate the association quality of responses—prioritizing depth, coherence, and insight over mere correctness. With regards to open-ended responses with multiple valid potential answers, our regular judge engine assess the internal consistency and reasoning quality by the cascading scoring rubric:

- **4 points:** Accurate, logically consistent, and insightful, matching the reference answer's intellectual rigor.

- **3 points:** Shows reasonable understanding but lacks key insights or completeness.

- **2 points:** Somewhat relevant but lacks depth, is overly broad, or omits critical reasoning.

- **1 point:** Vague, uncertain, or incomplete, failing to provide meaningful reasoning.

- **0 points:** Contains factual errors or fabrications that undermine validity.

We refer to this scoring as the *Holistic Score* in the paper to distinguish it from the reasoning score introduced in Section 4.2. Based on the scoring rubic, we define the evaluation metrics: (1) *Score Rate (SR)*, the average score to all open-ended responses judged by the LLM to reflect the general performance. (2) *High Score Rate (HR)*, the proportion of responses with explanation that makes sense in terms of LLM's analysis. It specifically derives *HR-3*, the percentage of responses scoring not less than 3, and *HR-4*, the percentage of responses scoring 4 (consistent with the reference answer). (3) It is obvious that *HR-3* $\geq$ *HR-4*, and their difference $\triangle HR = HR\text{-}3 - HR\text{-}4$ implies the proportion of the "divergent thinking" results of LVLMs.

### 4.2 Process-Reward LLM-as-a-Judge Scoring

The regular scoring rule in Section 4.1 is outcome-based and fail to distinguish and analyze models that produce similar outcomes through divergent thinking with different reasoning paths. Drawing inspiration from process reward models [85, 51], which qualify each intermediate reasoning step based on its potential to reach the correct outcome, we propose a customized process-reward LLM-as-a-Judge method (PR-Judge) to access each association reasoning step towards the final outcome connections, offering insights of reasoning process that outcome-based metrics cannot capture.

1. **Path Construction**: The LLM judge reformats model responses into association paths $P$ comprising sequential steps (or hops) $(s_1, s_2, ..., s_n)$.

2. **Stepwise Scoring Indicators**: Association reasoning step $t$ is accessed from three persepecitves:

   - **Reasonableness** ($R_t$): Reasoning fluency, the cognitive fluidity and logical coherence of the associative transition, reflecting the plausibility that leads to the outcome.
   - **Distinctiveness** ($D_t$): The distinctiveness of concept boundaries. Lower value indicates the negative effect due to vague or overly general associative connections.
   - **Knowledgeability** ($K_t$): The level of detail and development of the idea relevant with domain knowledge manifested in the step.

   These stepwise indicators are inspired from Guilford's Alternate Uses [82] that reflects the divergent thinking behaviors of human. $R_t$ and $D_t$ are scalar values in $[0, 1]$ while $K_t$ is binary in (0 or 1).

3. **Stepwise Association Quality and Path Scoring**: With regards to the indicators, the association quality per step $s_t$ is calculated as:

$$s_t = \alpha R_t D_t + (1 - \alpha) K_t, \tag{1}$$

   then overall *Reasoning Score* of each reasoning path is:

$$S_r = \sum_{t=1}^{n} s_t \delta^t. \tag{2}$$

   Among them, $\alpha$ balances internal reasoning coherence $R_t D_t$ against knowledge $K_t$; $\delta$ serves as a cognitive decay factor resembling the spirit of self-supervised process reward model [85], inherently favoring efficient and precise reasoning paths.

This structured evaluation framework enables a comprehensive assessment of associative reasoning quality.

| | Remote-Item Association Task | | | | In-Context Association Task | | | |
|---|---|---|---|---|---|---|---|---|
| **Model** | SR(%) | HR-4(%) | HR-3(%) | △HR(%) | SR(%) | HR-4(%) | HR-3(%) | △HR(%) |
| **Proprietary LVLMs** | | | | | | | | |
| Claude-3.5-Sonnet | 49.38 | 9.26 | 25.17 | 15.91 | 49.35 | 3.97 | 23.27 | 19.3 |
| Gemini-1.5-Flash | 55.86 | 7.88 | 22.91 | 15.03 | 51.05 | 1.38 | 14.51 | 13.13 |
| Gemini-1.5-Pro | 45.34 | 8.95 | 20.97 | 12.02 | 42.16 | 2.45 | 11.05 | 8.60 |
| Qwen-VL-Max | 44.16 | 6.32 | 20.43 | 14.11 | 49.32 | 4.08 | 25.07 | 20.99 |
| Qwen-VL-Plus | 42.56 | 4.03 | 17.82 | 13.79 | 44.79 | 1.24 | 16.57 | 15.33 |
| Gemini-2.0-Flash-Thinking-Exp | 59.11 | 17.73 | 36.60 | **18.87** | 61.42 | 9.74 | 37.88 | 28.14 |
| Gemini-2.5-Pro-Preview | 60.05 | **23.89** | **41.75** | 17.86 | **63.09** | **12.85** | **41.15** | **28.30** |
| o4-mini | **60.33** | 19.86 | 37.89 | 18.03 | 61.55 | 10.24 | 36.60 | 26.36 |
| GPT-4o | 59.72 | 10.89 | 28.83 | 17.94 | 58.26 | 6.27 | 29.62 | 23.35 |
| **OpenSource LVLMs** | | | | | | | | |
| GLM-4V | 26.92 | 0.49 | 4.73 | 4.24 | 43.63 | 0.20 | 3.67 | 3.47 |
| InternVL-Chat-V1-2 | 36.41 | 3.52 | 16.02 | 12.5 | 34.30 | 0.62 | 9.59 | 8.97 |
| InternLM-XComposer2.5-7B | 50.21 | 2.21 | 14.39 | 12.18 | 44.87 | **1.41** | 18.18 | 16.77 |
| VILA1.5 | 46.72 | 2.45 | 15.38 | 12.93 | 44.46 | 1.27 | 14.93 | 13.66 |
| Yi-VL-34B | 45.25 | 4.97 | 19.63 | 14.66 | **54.39** | 1.30 | **19.53** | **18.23** |
| Qwen2.5-VL-7B-Instruct | **52.28** | **5.35** | **20.36** | **15.00** | 53.50 | 1.08 | 16.62 | 15.54 |
| Kimi-VL-A3B-Instruct | 48.41 | 5.14 | 16.43 | 11.30 | 48.96 | 0.94 | 14.17 | 13.22 |
| Human* | **61.88** | **22.84** | **48.97** | **26.13** | **68.69** | **31.65** | **61.47** | **29.82** |

Table 2: Performance of models and human on the RIA and ICA tasks judged by gpt-4o-2024-08-06, with metrics including the holistic score rate (SR), high score rate (HR-4 , HR-3, and △HR) derived from regular LLM-as-a-Judge. *The human baseline is based on the sampled data items.

# 5 Experiments and Analysis

## 5.1 Settings

**LVLM Baselines.** We evaluated both proprietary and open-source VLMs under zero-shot conditions with default temperature. Proprietary models[2] include GPT-4 Omni [69], o4-mini [70], Gemini-1.5-Pro [79], Gemini-1.5-Flash [79], Gemini-2.5-Pro-Preview [27], Gemini-2.0-Flash-Thinking-Experimental [26], Claude-3.5-Sonnet [6], Qwen-VL-Max [8], Qwen-VL-Plus [8], while open-source models consist of GLM-4V [34], Yi-VL-34B [93], InternVL-Chat-V1-2 [20], VILA1.5 [52], InternLM-XComposer2.5-7B [99], Qwen2.5-VL-7B-Instruct [9] and Kimi-VL-A3B-Instruct [80]. Experiments for locally deployed models were conducted using 80 GB NVIDIA A800 GPUs.

**Human Baseline.** The study included 24 undergraduate and graduate students from diverse academic fields at a comprehensive university, selected for their cognitive skills appropriate for associative reasoning. We utilized 485 RIA and 436 ICA questions, grounded in widely accessible knowledge. Participants undertook the open-ended questions in a relaxed, non-evaluative atmosphere. Each addressed a subset of under 40 questions to ensure focus and prevent task-induced fatigue.

**Judge Engine.** We employ GPT-4o (gpt-4o-2024-08-06) and DeepSeek-V3 [28] as the mixed basic LLM-as-a-Judge engine for scoring. The former is excluded to evaluate its LVLM variant to ensure the fairness and prevent self-enhancement bias.

## 5.2 Outcome Evaluation of Association Reasoning

A comparison of different VLMs using the MM-OPERA is detailed in Table 2. Analyses across various dimensions are in Appendix B.1. Our key findings are:

**LVLMs Far Below Humans in Association Reasoning.** MM-OPERA reveals the formidable challenges of associative reasoning for current LVLMs. While latest models like o4-mini and latest Gemini models show improved performance, with SR approaching the human baseline, they still fall short in achieving high-quality associations. For instance, on the RIA task, o4-mini achieves an HR-4 of 19.86% compared to humans' 22.84%, and on the ICA task, Gemini-2.5-Pro-Preview reaches an HR-4 of 12.85% against humans' 31.65%, which demonstrates that sophisticated associative

---

[2]The model versions are: gpt-4o, o4-mini, gemini-1.5-pro-001, gemini-1.5-flash-001, gemini-2.5-pro-preview-05-06, gemini-2.0-flash-thinking-exp-01-21, claude-3-5-sonnet-20240620, qwen-vl-max-0809, qwen-vl-plus-0809.

reasoning remains at the cutting edge of LVLM capabilities. The fact that human performance is far from perfect is consistent with decades of psychometric research like [2] (the average human performance on the Remote Associates Test was 34.2% and it was rather low.) The performance gap between human and LVLMs is therefore an important scientific finding about the current state of AI. Case studies in Appendix C illuminate key limitations, such as **cross-domain knowledge retrieval deficiencies** and **perceptual misalignments**.

**Creativity Gap in Divergent Thinking.** The $\triangle$HR metric highlights divergent thinking, with most models scoring 12%–20%, showing their ability to generate reasonable yet non-optimal associations. Latest Gemini models lead among LVLMs (18.87% and 28.30% in two tasks), but humans outperform with both higher $\triangle$HR (26.13% and 29.82%) and HR-3 scores, demonstrating a superior balance of creativity and accuracy—an area where LVLMs remain limited.

**ICA: Dual Challenge of Pattern Abstraction and Transfer.** Most models perform better on RIA than ICA, highlighting the challenges of pattern-based associative reasoning. ICA requires not only connecting concepts but also abstracting and transferring these patterns to new contexts—a complex process demanding advanced meta-reasoning. Notably, some models such as latest Gemini models, Yi-VL-34B and GLM-4V outperform on ICA compared to RIA, suggesting that certain architectures excel in specific associative reasoning tasks. These distinctions may stem from more effective pattern extraction or transfer mechanisms, warranting further investigation.

**Conservative Reasoning vs. Associative Flexibility.** Analysis shows an inverse correlation between model constraints and associative abilities. Gemini-1.5-Flash (55.86% SR on RIA), optimized for speed, outperforms Gemini-1.5-Pro (45.34% SR), despite Pro's larger size and focus on detailed reasoning. Examination of 500 random RIA samples (Figure 3) shows Pro's conservative behavior to reason the high-rate association, prioritizing factuality and ethics, led to 1 point scores on nearly 20% of RIA questions due to conservative responses like "unrelated", versus Flash's <10%. Flash tended to offer superficial connections where Pro declined. Thus, factuality checks and ethical considerations, while improving reliability for complex tasks, can limit performance on creative association.

## 5.3 Process Evaluation of Association Reasoning

To deeply understand LVLMs' associative reasoning capabilities, we conducted fine-grained analysis using our Process-Reward LLM-as-a-Judge (PR-Judge) on 500 samples each from RIA and ICA datasets. We evaluated 9 models with $\alpha = 0.9$ and $\delta = 0.9$ employing both GPT-4o and Deepseek-V3 as the judges, averaging their results for final analysis. This dual-judge approach mitigates self-enhancement bias, as Deepseek-V3 provides an independent perspective with its distinct architecture and specialized mixture-of-experts training methodology. Though judges showed slight variance in scoring ranges (Deepseek-V3 trending higher), the self-enhancement bias of GPT-4o and the impact on comparative rankings remained minimal (Appendix B.2).

**Process Evaluation of Association: Complexity Matters.** GPT-4o demonstrates superior performance on both tasks, achieving the highest scores across all metrics (see Table 3). All models achieved average reasoning scores above 1.1 on RIA, but these scores dropped below 0.7 on ICA. Figure 3's Reasoning Score and Hop Count Distribution reveal that RIA responses exhibit richer reasoning structures, primarily centered at higher reasoning scores and 2-hop paths. In contrast, ICA tasks generate a substantial proportion of low scores and 0-hop responses, often reflecting insufficient logical structure or vague associative connections, thus highlighting ICA's greater difficulty and complexity.

**Plausible Links vs. Knowledge-Grounded Distinctiveness.** Figure 3's distributions of Reasonableness, Distinctiveness, and Knowledgeability reveal a critical limitation in LVLMs' associative reasoning: they struggle to move beyond plausible connections to achieve clear, knowledge-grounded understanding. While performing adequately on Reasonableness (50%–80% of RIA responses scoring above 75%), models significantly fall short in distinctiveness (less than half above 75%). Knowledgeability scores, though generally higher, still show a shortcoming in deep knowledge integration. This is reflected by the concentration of holistic scores at a mediocre "2" (Figure 3's Holistic Score Distribution), indicating superficial relevance and lack of depth. Thus, LVLMs can establish plausible connections, but lack the clear conceptualization and comprehensive knowledge integration required for truly sophisticated associative thinking.

| | RIA | | ICA | |
|---|---|---|---|---|
| Model | Holistic SR(%) | Avg. Reasoning Score | Holistic SR(%) | Avg. Reasoning Score |
| Claude-3.5-Sonnet | 58.15 | 1.4148 | 49.28 | 0.5099 |
| Gemini-1.5-Flash | 61.95 | 1.4193 | 52.95 | 0.3746 |
| Gemini-1.5-Pro | 58.35 | 1.3805 | 41.38 | 0.2208 |
| Qwen-VL-Max | 54.45 | 1.3160 | 50.375 | 0.6346 |
| Qwen-VL-Plus | 56.20 | 1.2362 | 47.68 | 0.4901 |
| GPT-4o | **67.78** | **1.6068** | **59.70** | **0.6396** |
| InternLM-XComposer2.5-7B | 54.38 | 1.1384 | 47.95 | 0.2144 |
| VILA1.5 | 55.73 | 1.1384 | 47.98 | 0.4191 |
| Yi-VL-34B | 58.43 | 1.2463 | 53.10 | 0.3567 |

Table 3: Holistic Score Rate (%) and average Reasoning Score of nine LVLMs on RIA and ICA tasks. Bold indicates best results, underlined indicates second-best results. Scores represent the average of evaluations by GPT-4o and Deepseek-V3 judges.

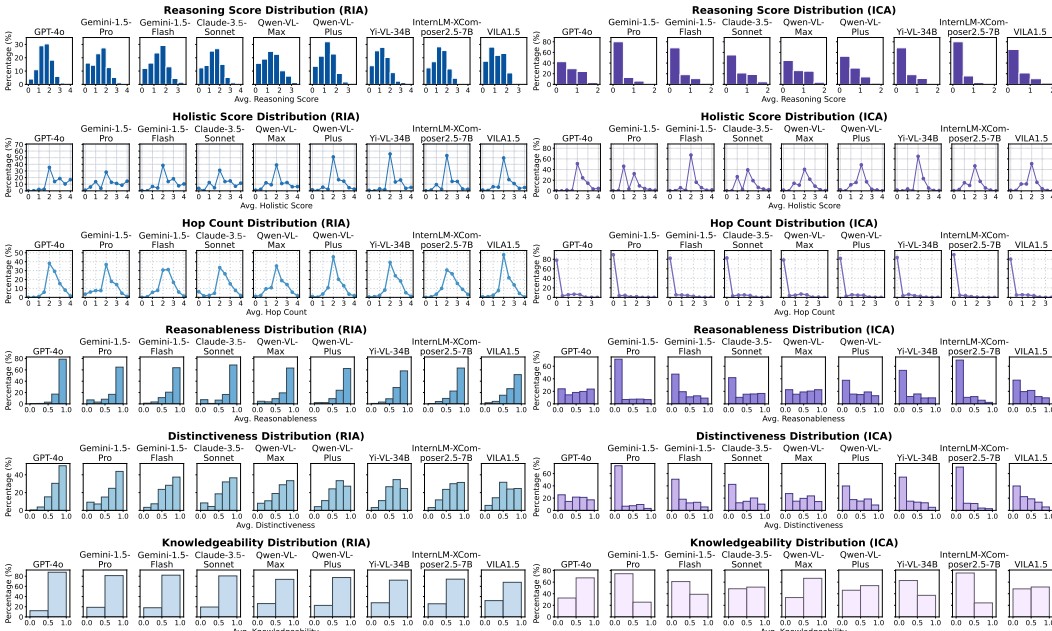

Figure 3: Fine-grained reasoning capability analysis of nine multimodal language models on RIA (left) and ICA tasks (right). From top to bottom: reasoning score distribution, holistic score distribution, reasoning path hop count distribution, Reasonableness distribution, Distinctiveness distribution, and Knowledgeability distribution. Each task includes 500 sampled questions, with results averaging evaluations from both GPT-4o and Deepseek-V3 judges.

## 5.4 Sensitivity Analysis of Task Instances

We conducted three sensitivity tests to assess score consistency and robustness. In the Multi-Image Substitution Test, we grouped multiple-image variants with identical concept pairs in RIA and measure score variability. In the Text-Image Substitution and Order Sensitivity tests, we randomly sampled 400 RIA instances and evaluated GPT-4o, Gemini-1.5-Pro, and Gemini-1.5-Flash, using original image-image pairs as the baseline.

**Multi-Image Substitution Test.** Results (detailed in Appendix B.3.1) reveal significant variability in how LVLMs handle different visual representations of identical concepts. GPT-4o demonstrates remarkable visual robustness with minimal score fluctuation, while some models show substantial performance variations across concept-identical images. This indicates most current LVLMs remain sensitive to surface-level visual features rather than forming robust conceptual representations, highlighting a critical gap between contemporary architectures and true concept-level associative reasoning.

**Text-Image Substitution Test.** We evaluated cross-modal generalization by replacing images with text descriptions and comparing scores across conditions. Appendix B.3.2 suggests GPT-4o's

reliance on nuanced visual cues that text descriptions cannot fully capture, while Gemini models demonstrate stronger text-equivalence in their reasoning, potentially processing visual information through language-like internal representations. These findings highlight how different architectural approaches influence cross-modal generalization in associative reasoning tasks. Additionally, the observations also showed that LVLMs struggle more with processing the raw visual input than with reasoning from a 'perfect' text description. It demonstrated how visually challenging our benchmark is.

**Order Sensitivity Test.** We examined the model's sensitivity to input order by reversing the image sequence. Appendix B.3.3 suggests that while GPT-4o processes image pairs in a more commutative manner, treating both ordering equally, Gemini models, particularly Gemini-1.5-Pro, appear to apply asymmetric reasoning processes that may prioritize the first image as context and the second as the target for association, highlighting architectural differences in how models approach bimodal associative reasoning.

### 5.5  LLM-as-a-Judge Strategy Validation for Reliable Evaluation

Full details of our LLM-as-a-Judge framework validation are in Appendix B.4.

**Bias Analysis.** We addressed **verbosity and position biases** [101]. Excluding short 1-point responses, the Pearson correlation between response length and scores was 0.376 for regular scoring and 0.291 for PR-Judge, indicating minimal verbosity bias. Permutation tests shuffling answer order showed mean score differences below 0.1 (regular) and 0.16 (PR-Judge), confirming negligible position bias.

**Alignment with Human Judgment.** We compared 300 sampled GPT-4o judgments with 8 human evaluators, yielding an average score difference of 0.077, with 78.33% perfect matches and no discrepancies exceeding 1 point. For PR-Judge, we evaluated 200 reasoning paths scored by 8 domain-expert judges on Reasonableness, Distinctiveness, and Knowledgeability. The average score difference was 0.1961, with 81% differing by less than 0.20 and none exceeding 0.60. Correlations were strong: $r = 0.72$ for Reasonableness, $r = 0.68$ for Distinctiveness, and 83.5% accuracy for Knowledgeability (Cohen's Kappa = 0.65), demonstrating robust alignment with human judgment.

**Effectiveness of Process-Reward LLM-as-a-Judge.** We compared it with outcome-based regular scoring on 100 paths. While outcome-based methods gave similar scores (e.g., 4) to correct answers, PR-Judge distinguished reasoning quality (e.g., 1.3 vs. 1.8), offering a more nuanced evaluation.

## 6  Conclusion

MM-OPERA introduces a novel framework for evaluating LVLM's association reasoning through open-ended tasks without predefined constraints. Drawing from cognitive psychology, it addresses traditional limitations while capturing diverse aspects of associative thinking. Results reveal that top LVLMs fail to achieve human performance, exposing task-specific patterns and a distinctiveness gap in robust conceptual reasoning. These insights underscore current limitations and provide direction for advancing human-like reasoning models.

## Acknowledgments and Disclosure of Funding

We would like to express our sincere gratitude to the numerous volunteers for their invaluable contributions to the development of the MM-OPERA benchmark. Their diligent efforts in data collection and quality control were crucial to the success of this work. We would especially like to thank Xiaotong Fu, Dawei Zheng, Hanbo Zhang, Chao Tan, Jinfeng Liu, Jiale Wang, Qinglin Zeng. We are also grateful to Zechuan Chen, Quanlong Guan, Xinghe Cheng, and Jialong Xue for their significant help in organizing the human baseline evaluation.

The research was supported in part by Guangdong S&T Programme (Grant No. 2024B0101010003); the Open research fund of Pengcheng Laboratory (Grant 2025KF1B0050); the Major Key Project of PCL (Grant Nos. PCL2024A04, PCL2025A02); the National Natural Science Foundation of China (NSFC) (Grant Nos. 62206110, 62176103, 62377208, 62572498, and 62276114); and the Science and Technology Planning Project of Guangzhou (Grant Nos. 2024A04J9896, 2025A03J3565).

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

## Technical Appendices

- Section A: More Benchmark Details
- Section B: Supplementary Results and Analysis
- Section C: Case Study 1: Why Do Models Perform Poorly?
- Section D: Case Study 2: Success Cases
- Section E: Test and Evaluation Prompts
- Section F: Limitations and Broader Impacts

## A  More Benchmark Details

### A.1  Association Path

We define three types of association paths to systematically represent different patterns of associative reasoning.

**Type 1: Sequential Association**

- Structure: $A \rightarrow X_1 \rightarrow X_2 \rightarrow B$
- Format:

$$\text{Predicate1}(A, X_1) \text{ and Predicate2}(X_1, X_2)$$
$$\text{Predicate3}(X_2, B)$$
$$A \rightarrow X_1 \rightarrow X_2 \rightarrow B$$

**Type 2: Convergent Association**

- Structure: $A \rightarrow X_1 \rightarrow X_2$ and $B \rightarrow X_2$
- Format:

$$\text{Predicate1}(A, X_1) \text{ and Predicate2}(X_1, X_2)$$
$$\text{Predicate3}(B, X_2)$$
$$A \rightarrow X_1 \rightarrow X_2 \text{ and } B \rightarrow X_2$$

**Type 3: Metaphorical Association**

- Structure: $A \wedge B \rightarrow X$
- Format: $A \wedge B \rightarrow X$

**Notation Conventions**: Entities and predicates follow PascalCase naming convention. The symbol 'and' connects separate relational clauses, while '$\wedge$' represents logical conjunction between entities. Each arrow ($\rightarrow$) represents one associative hop.

While the examples above demonstrate paths with one or three hops, the actual number of intermediate nodes ($X_i$) and associative steps may vary depending on the complexity of the reasoning process.

### A.2  Hierarchical Association Annotation

We develop a hierarchical annotation framework to systematically evaluate multimodal associative reasoning abilities. The framework consists of three levels that progress from basic perception to complex conceptual reasoning:

**Level-1 (L-1)** divides associative abilities into two fundamental categories:

- *Perception*: Processes immediate sensory inputs, focusing on visual understanding and interpretation
- *Conception*: Handles abstract, knowledge-driven associations requiring higher-order cognitive processing

**Level-2 (L-2)** further refines these categories into six dimensions:

- Under *Perception*: Recognition, Context, and Interaction
- Under *Conception*: Logic, Semantic, and Reasoning

**Level-3 (L-3)** provides the most granular classification with thirteen specific dimensions. Each dimension captures a distinct aspect of associative reasoning. Table 4 presents detailed definitions for each dimension.

Table 4: Detailed Definitions of Hierarchical Association Dimensions

| L-1 | L-2 | L-3 |
|---|---|---|
| **Perception** | **Recognition** | **Visual Similarity** 
 Associations based on visual features like shape, color, texture, and appearance. |
| | | **Semantic Object** 
 High-level semantic recognition of objects, including fine-grained identification in specific contexts. |
| | **Context** | **Contextual Sensory Cues** 
 Perceptual associations based on visual details like tone, lighting, and spatial layout. |
| | | **Scene Contextualization** 
 Understanding of overall scene context, including atmosphere and purpose. |
| | | **Abstract Interpretation** 
 Recognition of abstract concepts and symbolic patterns. |
| | **Interaction** | **Social Insight** 
 Understanding emotions and interactions between people in visual scenes. |
| | | **Relational Perception** 
 Comprehension of spatial and logical relationships between objects. |
| **Conception** | **Logic** | **Functional Links** 
 Associations based on functional relationships between concepts. |
| | | **Causal Connections** 
 Associations based on cause-and-effect relationships. |
| | **Semantic** | **Thematic Links** 
 Associations within the same theme or context. |
| | | **Cultural Reference** 
 Associations based on cultural knowledge and specific contexts. |
| | **Reasoning** | **Hierarchical Association** 
 Vertical associations between abstract and concrete concepts. |
| | | **Analogical Reasoning** 
 Associations based on structural, feature, or pattern similarities. |

This hierarchical framework enables systematic evaluation of LVLMs' associative abilities across different cognitive levels, from basic sensory processing to sophisticated abstract reasoning. The progression from L-1 to L-3 mirrors human cognitive development and provides a comprehensive structure for analyzing multimodal understanding capabilities.

## A.3   Data Sources

The MM-OPERA-Bench dataset, consisting of images, reference answers, and fine-grained annotations, was manually curated by a group of volunteers. Of the total data, 33.35% of the questions and reference answers were sourced from the RAT [64], while 4.01% of the images, questions, and reference answers were sourced from the LI-RAT [11] datasets for human psychometric testing. The remaining images were sourced from the Internet, and all fine-grained annotations were manually constructed and revised to ensure consistency and accuracy.

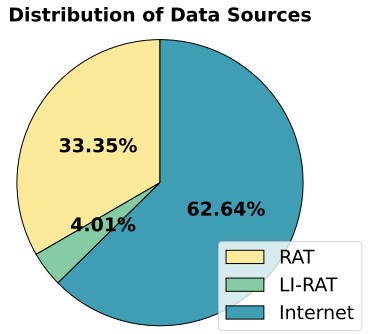

Figure 4: Distribution of data sources.

## A.4 Data Collection and Curation Protocol

The MM-OPERA dataset, encompassing images, reference answers (including reasoning paths), and multifaceted annotations, was meticulously curated through a multi-stage, volunteer-driven process, adhering to ethical guidelines and scientific rigor.

### A.4.1 Data Collection

Volunteers, primarily undergraduate and graduate students from diverse disciplines (STEM, humanities, social sciences, arts), were recruited via university channels to leverage their strong cognitive abilities and varied perspectives, enriching the dataset creation process with a broad range of knowledge and insights. They received training on project goals, task details, data privacy considerations (anonymized contributions), and time commitment. Guidelines covered associative attribute definitions, example generation, image sourcing (avoiding unsafe or inappropriate content), and annotation consistency. Participation was voluntary, with contributors acknowledged.

A core research team created 10–20 high-quality seed instances for each of the 13 Level-3 (L-3) associative attributes as exemplars. Volunteers expanded the dataset by sourcing images from public repositories (e.g., Wikimedia Commons, public domain archives) and adapting items from psychometric tests (e.g., RAT and LI-RAT). They were trained to exclude images depicting illegal, violent, or offensive content, using safe search filters and careful judgment. Sourced items underwent manual revision to ensure appropriateness, clear associative links, plausible reasoning paths, and diverse associations beyond original tests. Volunteers were also guided to create instances reflecting cultural contexts, linguistic nuances (English-based items testing concepts across 15 linguistic backgrounds), and thematic domains (22 topic domains to avoid biases). A tracking system ensured balanced coverage, prompting targeted collection if gaps were identified.

### A.4.2 Quality Control

A multi-layered quality control process ensured accuracy, clarity, challenge, and safety of the MM-OPERA benchmark. Each instance underwent initial screening by the core team for guideline adherence, including checks for inappropriate images. A two-stage peer review followed: (1) **Cross-Review**: Two uninvolved volunteers assessed clarity, relevance, reasoning plausibility, formatting, and image safety, providing revision feedback. (2) **Expert Review**: Core researchers evaluated conceptual soundness, difficulty, biases, and safety, discarding or revising problematic items. Five core team members then assessed instance difficulty (Easy, Medium, Hard, Very Hard) based on association remoteness, reasoning complexity, and cue subtlety. Consensus was reached through discussion. Approximately 5% of instances (too trivial or obscure) were excluded to ensure meaningful challenges. Feedback from quality control refined guidelines and training.

Crucially, what sets our validation apart is the **structured Association Reasoning Path** included with every instance thanks to our Process-Reward LLM-as-a-Judge method. Reviewers validated the entire step-by-step logical chain, ensuring that the association is not only plausible but also coherently and correctly reasoned. This traceable reasoning provides a far more robust and objective measure of correctness than benchmarks with only a final label, significantly mitigating the risk of flawed or ambiguous examples.

The final dataset includes only instances passing all review and calibration stages, ensuring a high-quality, diverse, challenging, and safe benchmark for evaluating associative reasoning in LVLMs.

# B    Supplementary Results and Analysis

## B.1    Multi-dimensional Analysis

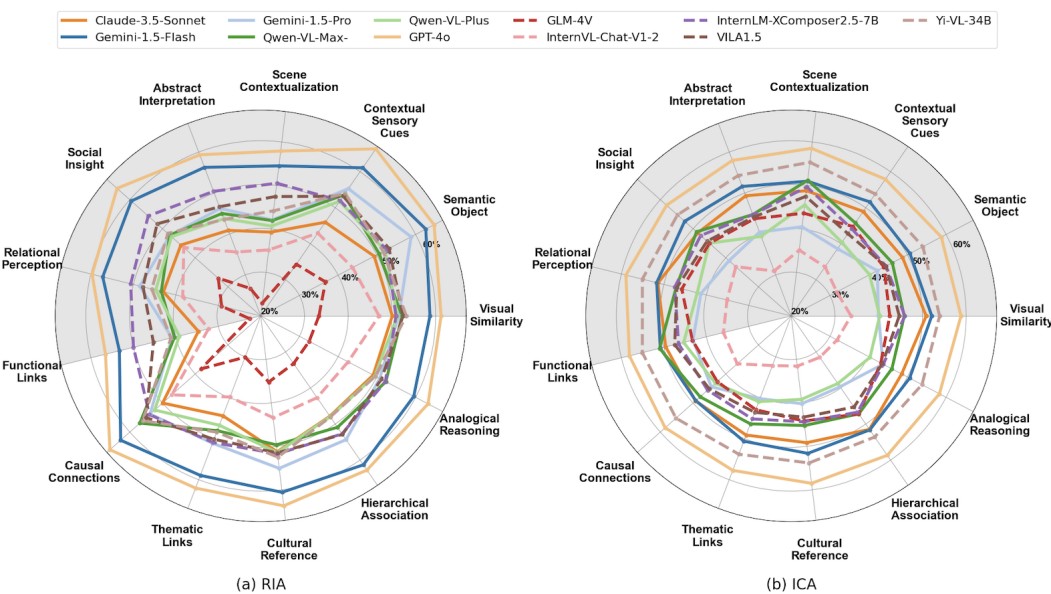

Figure 5: Comparison of Model Performance in RIA and ICA across Different Conceptual (white background) and Perceptual (gray background) Dimensions. The radar charts illustrate the capabilities of various LVLMs in handling tasks related to relational perception, social insight, causal connections, abstract interpretation, and other cognitive functions. The left chart (RIA) exhibits greater variability in model performance, while the right chart (ICA) shows more consistent trends across models.

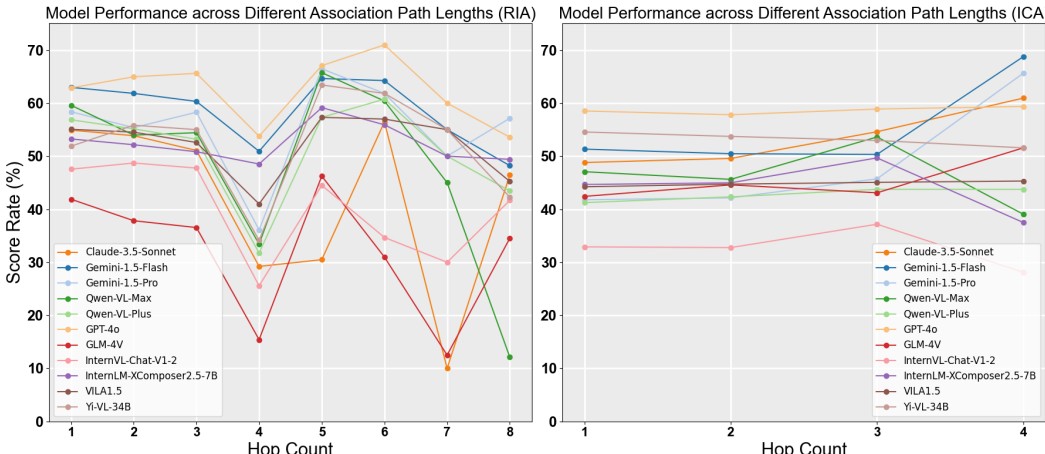

Figure 6: Model Performance Across Different Association Path Lengths in RIA and ICA tasks. The line graphs illustrate the score rates (%) of various LVLMs as the number of association path "hops" increases. The left chart represents RIA results, showing notable fluctuations in performance across different hop counts. The right chart represents ICA results, where models generally display more stable trends. This analysis highlights how different models handle varying levels of associative complexity.

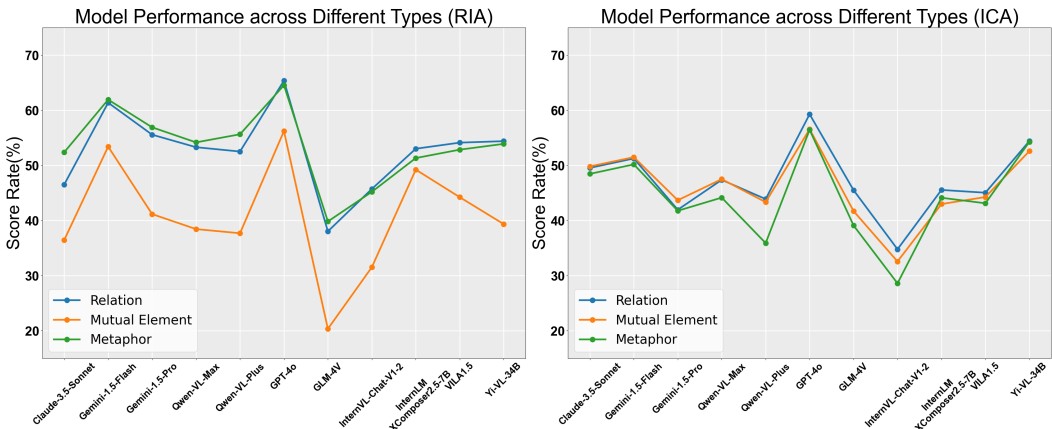

Figure 7: The figure presents the performance of various multimodal large models across different reasoning types in the RIA (left) and ICA (right) tasks. The three reasoning types—Relation, Mutual Element, and Metaphor—are represented by different colored lines. The vertical axis indicates the score rate (%), while the horizontal axis lists different models. The results show varying performance trends across reasoning types and tasks, highlighting differences in model capabilities in handling relational, compositional, and metaphorical understanding.

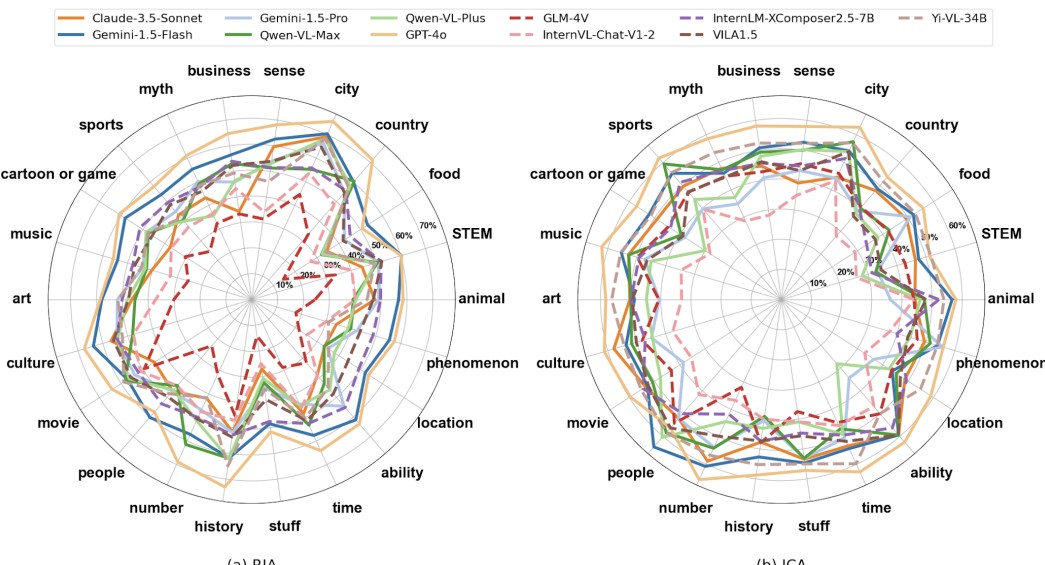

Figure 8: Radar Chart Comparison of Model Performance Across Domains in RIA and ICA tasks. The two radar charts display the performance of various LVLMs across different knowledge domains such as business, sports, music, STEM, history, and culture. The left chart (a) represents results from the RIA tasks, while the right chart (b) shows ICA results. The models exhibit varying performance across different domains, with some excelling in specific categories while struggling in others.

Multidimensional analysis reveals the complex landscape of associative reasoning capabilities in Large Vision Language Models (LVLMs). Most models perform better on RIA tasks compared to ICA tasks, with an average performance differential of approximately 5–7 percentage points. This suggests that identifying direct associations between unrelated items may be more tractable for current LVLMs than recognizing and extending associative patterns. The exception is Claude-3.5-Sonnet, which shows relatively consistent performance across both task types, indicating potentially more balanced associative reasoning capabilities. These findings underscore the multi-faceted nature of associative cognition and the importance of diverse task designs for comprehensive evaluation.

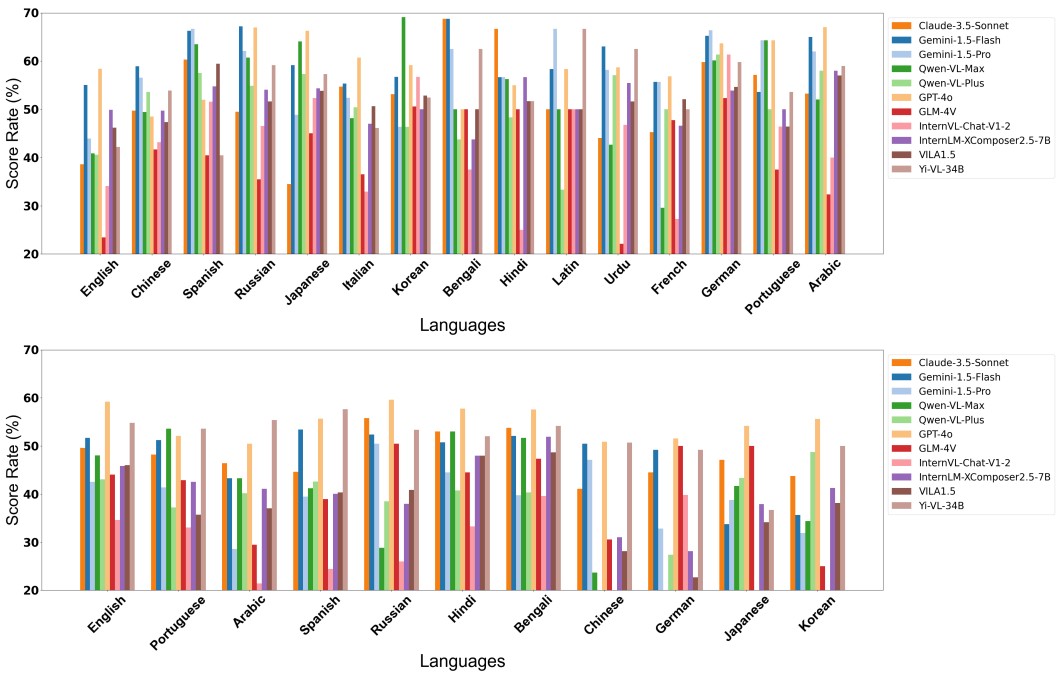

Figure 9: The figures compare the performance of different multimodal large models across multiple languages. The left figure represents the RIA tasks, while the right figure corresponds to the ICA tasks. The vertical axis indicates the score rate (%), and the horizontal axis lists various languages, including English, Chinese, Spanish, and others. The results highlight significant differences in model performance across tasks and languages, reflecting their varying capabilities in cross-linguistic understanding and reasoning.

**Reasoning Complexity and Cognitive Abilities.** Analysis of reasoning complexity reveals non-linear patterns in how models handle associative tasks. Figure 6 shows that while most models effectively manage simple 1–2 hop associations (with score rates around 50–60%), performance drops significantly for more complex 4-hop associations (29–53%). However, some models (e.g., GPT-4o, Gemini-1.5-Flash, and Qwen-VL-Plus) demonstrate relatively stable performance in very complex 5–8 hop associations, suggesting the emergence of new strategies in complex reasoning paths. This "complexity valley" phenomenon warrants further investigation as it may provide important insights into how LVLMs structure multi-step associative reasoning. Differences in perceptual and conceptual abilities are evident in Figure 5. "Semantic Object" in perceptual abilities and "Causal Connections" in conceptual abilities show stronger performance, while "Abstract Interpretation" and "Functional Links" remain challenging. Cross-task analysis indicates that models maintain consistent relative strengths across RIA and ICA tasks, but absolute performance levels are modulated by task demands, especially for perceptual abilities. This suggests that while underlying reasoning mechanisms remain stable, their expression is influenced by task requirements.

**Relationship Types.** Analyzing association types reveals distinctive performance patterns between Remote-Item Association (RIA) and In-Context Association (ICA) tasks. As shown in Figure 7, in RIA tasks, models demonstrate a pronounced hierarchy among association types, with Metaphor associations yielding the highest performance (52–65% for top performers), followed by Relation associations (46–65%), and Mutual Element associations showing the lowest scores (36–56%). This hierarchy is notably consistent across nearly all models. Interestingly, in ICA tasks, this performance stratification significantly diminishes, with much smaller performance gaps between association types. For instance, GPT-4o shows only a 2.82 percentage point difference between its highest (Relation: 59.29%) and lowest (Mutual Element: 56.59%) association type performance in ICA, compared to a 7.15 point gap in RIA. This convergence suggests that the contextual framework provided in ICA tasks may equalize the difficulty of recognizing different association types.

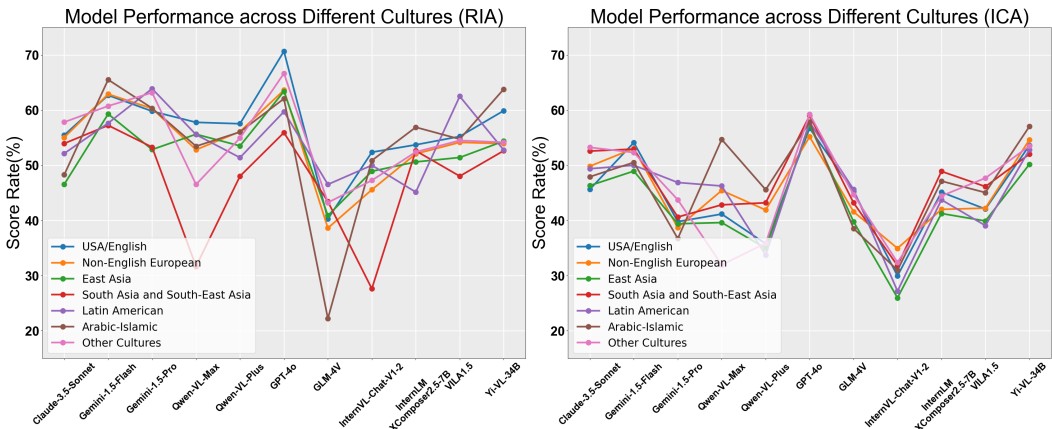

Figure 10: Comparison of Model Performance Across Different Cultures in RIA and ICA tasks. The line plots illustrate the score rates (%) of various LVLMs across cultural groups, including USA/English, Non-English European, East Asian, South Asian and South-East Asian, Latin American, Arabic-Islamic, and Other Cultures. The left graph represents RIA results, while the right graph shows ICA results

**Domain and Cultural Dimensions.** Domain knowledge differences are highly evident in model performance. As shown in Figure 8, urban-related associations consistently achieve higher performance (around 65–75% for top models), while everyday objects and food-related associations pose greater challenges (around 30–45%). These differences suggest inherent difficulties in forming associations within certain conceptual spaces. GPT-4o excels in history-related associations (73.07%), significantly outperforming other models, which may indicate superior historical knowledge representation or more effective temporal association retrieval mechanisms. Cultural background also significantly impacts model performance. Figure 10 reveals that most models show stronger associative reasoning when dealing with Western cultural references compared to East Asian, South Asian, or Arabic-Islamic contexts. In Figure 9, similar asymmetries are observed in language performance, with models generally performing better in Spanish, German, and Russian associations than in East Asian languages. These cultural and linguistic disparities may reflect imbalances in multilingual pretraining or fundamental differences in how associations manifest across different linguistic and cultural structures.

**Multilingual Capability.** The current language distribution (in Figure 2.e) is a principled design choice to robustly evaluate distinct aspects of associative reasoning. On one hand, English serves as the dominant language. Current LVLMs are primarily trained on English-centric data, so using English as the primary language offers a fair and stable baseline for evaluating core capabilities. On the other hand, to achieve a more comprehensive evaluation, we additionally include non-English samples, which serve as targeted probes for culturally nuanced associative phenomena. These include challenges such as linguistic wordplay (e.g., homophones or puns unique to a given language) and cultural knowledge (e.g., proverbs, historical references, or artistic expressions that require deep, language-specific world knowledge). The design of non-English samples requires the cultural context associated with each language, resulting in more complex construction logic and thus a more limited pool of suitable examples. This is why non-English samples follow a long-tail distribution.

To provide a more balanced view of multilingual capability, we report the harmonic mean ($H_{SR}$) of the score rate (SR) on English and non-English samples in Table 5. This metric mitigates the dominance of the larger English subset and is sensitive to performance disparities across languages. The $H_{SR}$ confirms that top models like Gemini-2.5-Pro and o4-mini demonstrate strong, balanced capabilities. This reinforces our conclusions while offering a more nuanced view of multilingual performance.

We also observe an interesting fact that in the RIA task, most models achieve a higher SR on non-English samples. We hypothesize this is because these instances often test specific, well-defined cultural knowledge (e.g., proverbs). A successful association relies on retrieving a precise factual link, which models with broad world knowledge can do effectively. Conversely, in the ICA task, many top

| | Remote-Item Association (RIA) Task | | | In-Context Association (ICA) Task | | |
|---|---|---|---|---|---|---|
| **Model** | $H_{SR}$ (%) | Eng SR (%) | Non-English SR (%) | $H_{SR}$ (%) | Eng SR (%) | Non-English SR (%) |
| **Proprietary LVLMs** | | | | | | |
| Claude-3.5-Sonnet | 52.92 | 48.37 | 58.41 | 48.94 | 49.66 | 48.25 |
| Gemini-1.5-Flash | 58.12 | 55.28 | 61.27 | 50.13 | 51.72 | 48.64 |
| Gemini-1.5-Pro | 49.70 | 44.17 | 56.80 | 41.63 | 42.55 | 40.74 |
| Qwen-VL-Max | 51.05 | 44.84 | 59.27 | 46.22 | 51.50 | 41.92 |
| Qwen-VL-Plus | 47.47 | 41.24 | 55.90 | 42.93 | 46.05 | 40.21 |
| Gemini-2.0-Flash-Thinking-Exp | 60.93 | 58.58 | 63.47 | 59.56 | 62.72 | 56.71 |
| Gemini-2.5-Pro-Preview | **63.53** | 58.96 | 68.87 | **61.05** | 64.53 | 57.94 |
| o4-mini | 62.39 | 59.74 | 65.28 | 59.90 | 62.72 | 57.32 |
| GPT-4o | 62.02 | 59.27 | 65.04 | 56.94 | 59.21 | 54.83 |
| **OpenSource LVLMs** | | | | | | |
| GLM-4V | 32.48 | 25.00 | 46.35 | 42.68 | 44.32 | 41.15 |
| InternVL-Chat-V1-2 | 40.72 | 35.14 | 48.41 | 31.08 | 36.37 | 27.14 |
| InternLM-XComposer2.5-7B | **51.36** | 49.98 | 52.82 | 43.52 | 45.83 | 41.44 |
| VILA1.5 | 49.10 | 46.14 | 52.45 | 42.17 | 46.00 | 38.92 |
| Yi-VL-34B | 50.26 | 43.98 | 58.65 | **53.51** | 55.04 | 52.06 |

Table 5: The harmonic mean of the score rate (SR) on English and non-English samples of various LVLMs on the Remote-Item Association (RIA) and In-Context Association (ICA) tasks. The best-performing model in each sub-category is highlighted in bold.

| | Remote-Item Association Task | | | | In-Context Association Task | | | |
|---|---|---|---|---|---|---|---|---|
| | Holistic SR (%) | | Avg. Reasoning Score | | Holistic SR (%) | | Avg. Reasoning Score | |
| **Model** | 4o | V3 | 4o | V3 | 4o | V3 | 4o | V3 |
| Claude-3.5-Sonnet | 56.20 | 60.10 | 1.2838 | 1.5457 | 50.4 | 48.15 | 0.4159 | 0.6039 |
| Gemini-1.5-Flash | 63.25 | 60.65 | 1.2701 | 1.5684 | 51.35 | 54.55 | 0.3507 | 0.3985 |
| Gemini-1.5-Pro | 56.90 | 59.80 | 1.2701 | 1.4908 | 40.55 | 42.20 | 0.1742 | 0.2674 |
| Qwen-VL-Max | 49.30 | 59.60 | 1.2587 | 1.3733 | 44.60 | 56.15 | 0.5584 | 0.7107 |
| Qwen-VL-Plus | 54.50 | 57.90 | 1.0511 | 1.4212 | 43.00 | 52.35 | 0.4011 | 0.5791 |
| GPT-4o | **67.80** | **67.75** | **1.4676** | **1.7459** | **59.80** | **59.60** | **0.5611** | **0.7180** |
| InternLM-XComposer2.5-7B | 52.80 | 55.95 | 1.1902 | 1.5345 | 45.15 | 50.75 | 0.1560 | 0.2727 |
| VILA1.5 | 54.25 | 57.20 | 0.9979 | 1.2788 | 43.30 | 52.65 | 0.3122 | 0.5259 |
| Yi-VL-34B | 57.65 | 59.20 | 1.1424 | 1.3502 | 52.85 | 53.35 | 0.3107 | 0.4027 |

Table 6: Performance comparison of models on 500 sampled Remote-Item Association and In-Context Association instances as evaluated by two judges (GPT-4o and Deepseek-V3). Metrics include holistic score rate (SR) and average reasoning score. The highest values for each metric are bolded, while the second-highest are underlined.

models perform better on English samples. ICA demands abstracting and transferring a relational pattern, a meta-reasoning skill. We posit this capability is more robustly developed for English, the primary language in pre-training data, where such abstract logical structures are more prevalent.

Note that although some items were designed with specific cultural or cultural contexts in mind, models may still generate alternative but valid associations without explicitly relying on those cues. Our open-ended evaluation rewards any well-justified reasoning path, so scores may reflect reasoning flexibility or general knowledge rather than direct cultural or linguistic awareness. Thus, performance differences cannot be solely attributed to language proficiency.

## B.2 Evaluation by Different Judges

The comparison between the two judges (GPT-4o and Deepseek-V3) highlights notable differences in their scoring tendencies (see Table 6). Judge Deepseek-V3 consistently assigns higher average reasoning scores across most models, suggesting a more lenient evaluation of reasoning depth or quality. However, for holistic score rate (SR), the differences are less consistent, with some models (e.g., Claude-3.5-Sonnet and Qwen-VL-Max) receiving higher SR from V3, while others (e.g., GPT-4o) show near parity between the two judges. These disparities underscore the importance of employing multiple evaluators to mitigate individual judgment bias and ensure robust evaluation of model performance.

The visualized score distributions in Figure 11 further highlight key differences in evaluation tendencies and scoring patterns between GPT-4o and Deepseek-V3 across different models and tasks.

**Calibration Pattern Comparison.** The visualized reasoning score distributions reveal distinctive evaluation tendencies between GPT-4o and Deepseek-V3 across models and tasks. While both evaluators maintain similar distribution shapes for each model, Deepseek-V3 consistently demonstrates a broader scoring range, particularly on RIA tasks where it occasionally assigns scores of 5–6 to top-performing models like GPT-4o and Claude-3.5-Sonnet—scores beyond GPT-4o's 0–4 scale. This suggests Deepseek-V3 employs a more granular assessment framework with higher ceiling effects. Additionally, GPT-4o shows more concentrated distributions with sharper peaks, while Deepseek-V3 exhibits more dispersed distributions, particularly in the mid-range scores. Despite these calibration differences, both evaluators converge on identifying the same relative performance hierarchy across models and consistently highlight the challenging nature of ICA tasks, where all models receive predominantly low scores (0-1) regardless of which system performs the evaluation.

**Evaluator Consistency and Minimal Self-Enhancement Bias.** The holistic score distributions reveal remarkable consistency between GPT-4o and Deepseek-V3 as evaluators, providing strong evidence against significant self-enhancement bias. Despite GPT-4o evaluating its own outputs, both evaluators produce strikingly similar distribution patterns across all models for both RIA and ICA tasks. Notably, GPT-4o does not disproportionately favor its own responses—its self-evaluation distribution closely mirrors Deepseek-V3's independent assessment, with both showing peaks at similar score points. This alignment is particularly evident in the ICA tasks, where both evaluators produce nearly identical bell-shaped distributions centered around scores 2–3 for all models. The consistency across different evaluators suggests that our evaluation framework successfully mitigates potential self-enhancement effects, reinforcing the reliability of our findings even when using an LLM to evaluate its own outputs. This methodological robustness strengthens confidence in the comparative analysis of associative reasoning capabilities across different LVLMs.

**Reliable Path Complexity Analysis.** Both GPT-4o and Deepseek-V3 extract nearly identical hop count distributions from the same model outputs, reinforcing the reliability of our path analysis methodology. This consistency in path complexity evaluation across different judges provides strong evidence that the observed patterns reflect genuine differences in associative reasoning strategies between tasks rather than evaluator bias.

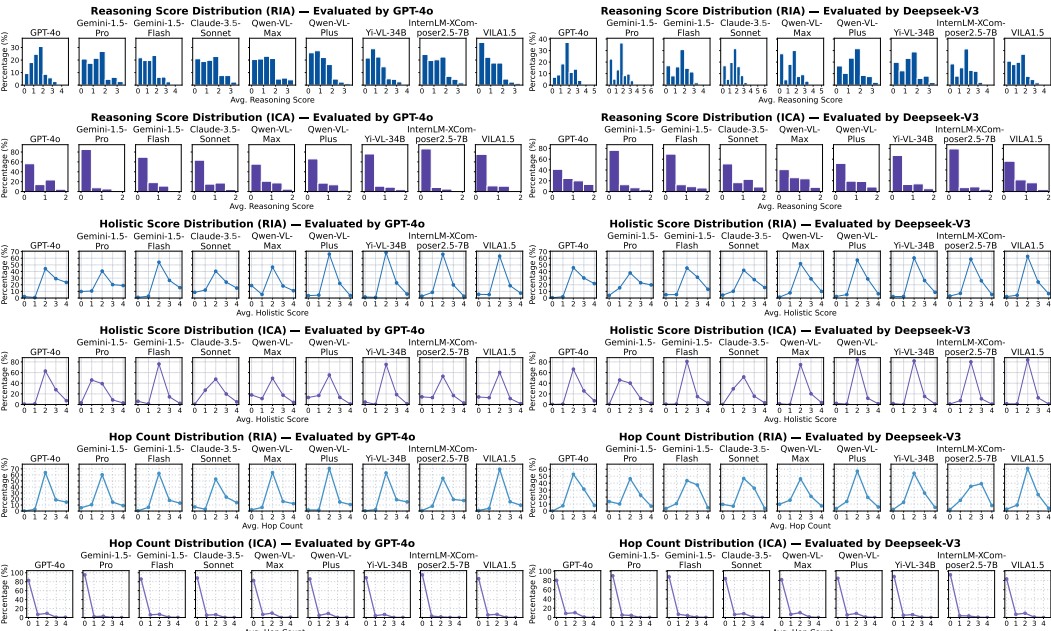

Figure 11: Fine-grained reasoning capability analysis of nine multimodal language models on RIA (blue) and ICA (purple) tasks judged by GPT-4o (left) and Deepseek-V3 (right). From top to bottom: reasoning score distribution, holistic score distribution, reasoning path hop count distribution. Each task includes 500 sampled questions.

## B.3 Sensitivity Test Results

| Model | IG Range↓ | IG SR↑ | IG SD↓ |
|---|---|---|---|
| Claude-3.5-Sonnet | 1.00 | 0.47 | 0.41 |
| Gemini-1.5-Flash | 0.68 | 0.55 | 0.27 |
| Gemini-1.5-Pro | 1.22 | 0.44 | 0.49 |
| Qwen-VL-Max | 0.99 | 0.43 | 0.38 |
| Qwen-VL-Plus | 1.06 | 0.41 | 0.41 |
| GPT-4o | **0.44** | **0.59** | **0.18** |
| GLM-4V | 0.79 | 0.25 | 0.30 |
| InternVL-Chat-V1-2 | 1.31 | 0.35 | 0.49 |
| InternLM-XComposer2.5-7B | 0.93 | 0.50 | 0.36 |
| VILA1.5 | 1.34 | 0.46 | 0.54 |
| Yi-VL-34B | 0.99 | 0.44 | 0.38 |

Table 7: Performance of models on the Multi-Image Substitution Test in RIA. We grouped multiple-image variants with identical concept pairs in RIA and measure score variability using *IG Range* (intra-group score range), *IG SR* (average intra-group score rate), and *IG SD* (intra-group standard deviation).

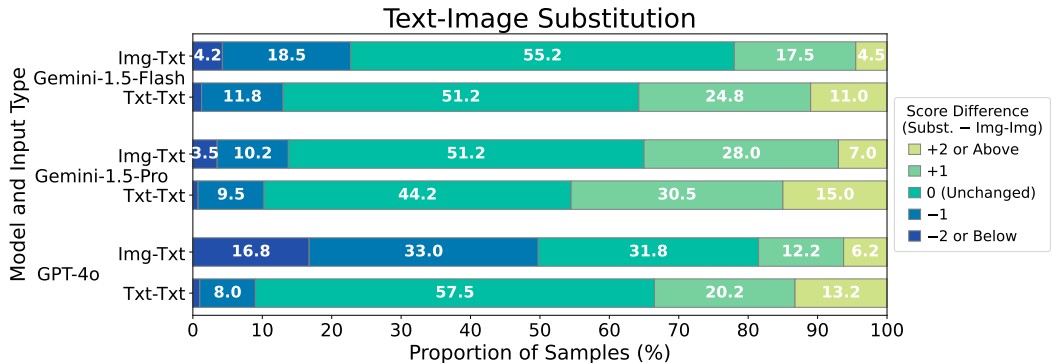

Figure 12: Score difference distribution for the Text-Image Substitution test across models in RIA. Bars show proportions of samples with varying score differences (substitution - original).

### B.3.1 Multi-Image Substitution Test

To assess robustness, we conducted sensitivity tests to measure how LVLMs' responses varied with different visual representations of the same concepts. Results in Table 7 revealed significant visual sensitivity across models. GPT-4o demonstrates exceptional consistency, showing the lowest intra-group score range (0.44) and standard deviation (0.18) while maintaining the highest score rate (0.59). In contrast, models like VILA1.5 and InternVL-Chat-V1-2 exhibit substantial variability (IG Ranges of 1.34 and 1.31, respectively) despite moderate performance, indicating that their associative reasoning is heavily influenced by specific visual features rather than robust concept understanding. This visual dependency suggests that most current LVLMs still associate at a surface feature level rather than at a deeper conceptual level—a critical limitation for real-world applications requiring consistent reasoning across variable visual inputs.

### B.3.2 Text-Image Substitution Test

Results in Figure 12 reveals distinct cross-modal generalization patterns across models. GPT-4o experiences the most significant performance drop when one image is replaced with the text description (Image-Text), with 49.8% of samples showing decreased scores (16.8% with severe drops of ≥2 points). Intriguingly, in the Txt-Txt setting, GPT-4o's performance is much more robust, with scores dropping for around 8.0% of samples. This reveals a deeper insight: GPT-4o may struggle with cross-modal fusion. It performs well when reasoning over vision-only or text-only inputs, but its performance falters when forced to integrate information from disparate modalities (Img-Txt).

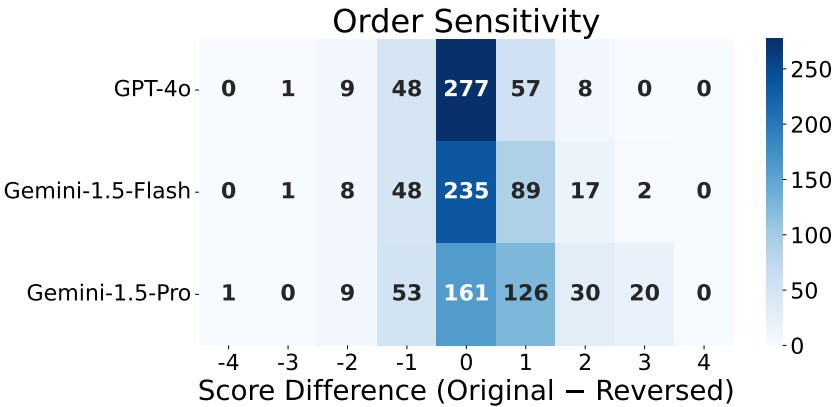

Figure 13: Order Sensitivity heatmap showing models versus score difference between original and reversed input order. Cell darkness indicates instance count.

In contrast, Gemini-1.5-Flash and Pro are far more resilient to modality substitution. In the Img-Txt setting, Gemini-1.5-Flash sees a performance drop in only 22.7% (4.2% + 18.5%) of cases, and scores are unchanged for a majority (55.2%) of samples. This suggests Gemini models may employ a different internal strategy, perhaps by more effectively converting visual inputs into a modality-agnostic, language-like representation.

These patterns also suggest fundamental differences in cross-modal processing strategies. GPT-4o appears more reliant on visual information for associative reasoning, extracting nuanced visual cues that text descriptions cannot fully capture. Meanwhile, Gemini models demonstrate stronger text-equivalence in their reasoning processes, suggesting they may process visual information by internally converting it to language-like representations. This finding highlights the importance of modality-specific evaluation when assessing LVLMs' associative reasoning capabilities.

These experiments have also demonstrated **how visually challenging the benchmark is**. When images are replaced with their text descriptions, the LVLMs' performances stayed the same or improved for over 87% of samples across all tested models (improved by $\geq 1$ for 33-46% samples and by $\geq 2$ for 11-15% samples). The observations can only be explained by LVLMs struggle more with processing the raw visual input than with reasoning from a 'perfect' text description. It leads to the evidence of the benchmark's visual challenge.

### B.3.3 Order Sensitivity Test

Results in Figure 13 reveals varying degrees of input order sensitivity across models. GPT-4o demonstrates the highest stability, with 277 of 400 instances (69.25%) showing no score change when input order is reversed. Gemini-1.5-Flash shows moderate consistency (58.75% unchanged), while Gemini-1.5-Pro exhibits notably lower order invariance (only 40.25% unchanged) with a significant rightward shift toward positive score differences, indicating better performance on the original order. This suggests that while GPT-4o processes image pairs in a more commutative manner, treating both ordering equally, Gemini models—particularly Gemini-1.5-Pro—appear to apply asymmetric reasoning processes that may prioritize the first image as context and the second as the target for association, highlighting architectural differences in how models approach bimodal associative reasoning.

### B.4 LLM-as-a-Judge Strategy Validation

For **human alignment** validation, We compared 300 randomly sampled scoring results of the model with those of 8 human evaluators. Furthermore, we analyzed potential biases in the LLM-as-a-Judge evaluation, focusing on **verbosity bias** by examining the correlation between response length and scores, and **position bias** through answer order permutation tests. Both analyses aimed to ensure an objective and consistent evaluation across models and human responses.

### B.4.1 Alignment with Human Judgment for Regular LLM-as-a-Judge Scoring

We compared 300 sampled GPT-4o's regular scoring results with those of human evaluators, finding an average score difference of 0.077. Notably, $78.33\%$ of the model's scores perfectly matched those of human judges, with $21.67\%$ of responses aligning within a 1-point difference. Critically, there were no instances of disagreement exceeding a 1-point margin, indicating strong calibration between our automated evaluation and human judgment. This high level of agreement demonstrates the reliability of our LLM-as-a-Judge framework for evaluating open-ended associative responses, effectively balancing the efficiency of automated assessment with the nuanced judgment characteristic of human evaluators. The absence of large scoring discrepancies further validates our approach as a robust proxy for human evaluation in this complex reasoning domain, addressing a key challenge in the assessment of open-ended multimodal tasks.

For the Process-Reward LLM-as-a-Judge (PR-Judge), we randomly selected 200 reasoning paths generated by the models and had them evaluated by 8 human judges with domain expertise. The human judges scored each reasoning step based on the same criteria used by the PR-Judge: Reasonableness $R_t$, Distinctiveness $D_t$, and Knowledgeability $K_t$. The overall reasoning score $S_r$ for each path was then calculated. Our results show that the average reasoning score difference between the human judges and the PR-Judge (GPT-4o) was 0.1961. Specifically, $81\%$ of the paths received scores differed by no more than 0.20 from the PR-Judge and human judges, while $16\%$ differed by no more than 0.50 points, and none had a difference more than 0.60, indicating a high level of agreement between the automated and human evaluations. We also observed strong positive correlations between the PR-Judge's scores and the average human scores: Pearson's $r = 0.72$ for Reasonableness, $r = 0.68$ for Distinctiveness. For the binary Knowledgeability indicator, the PR-Judge achieved an accuracy of $83.5\%$ (Cohen's Kappa = 0.65) compared to the majority human vote. These findings suggest that the PR-Judge effectively captures human-like nuances in assessing the quality of individual reasoning steps.

### B.4.2 Effectiveness of Process-Reward LLM-as-a-Judge

To justify the introduction of the Process-Reward LLM-as-a-Judge, we compared its performance with a traditional outcome-based scoring method using the same 100 reasoning paths. We found that the outcome-based method often assigned similar scores to models that produced correct outcomes but through different reasoning processes. For instance, two models might both receive a score of 4 based on their final answers, but the Process-Reward method revealed differences in their reasoning quality, with one model scoring 1.3 and the other 1.8, reflecting the latter's superior reasoning process. This demonstrates that the Process-Reward approach provides a more nuanced evaluation of reasoning quality compared to traditional methods.

### B.4.3 Bias Analysis

We investigated potential biases in LLM-based evaluation.

**Verbosity bias.** Since 1-point responses are significantly shorter due to their vague or uncertain nature, we excluded them and compared the correlation between response length and performance. Our analysis yielded a Pearson Correlation coefficient of 0.376 for regular scoring and 0.291 for PR-Judge. This moderate positive correlation is acceptable, as high-quality responses often require more detailed explanations. The correlation is not strong enough to suggest that the LLM judge is primarily influenced by response length rather than content quality.

**Position bias.** We performed permutation tests on 500 samples each on RIA and ICA tasks by randomly shuffling the order of the standard and model-generated answers in the judging prompt. The results showed no systematic advantage for any position, with mean score differences across permutations averaging 0.0871 for regular scoring and 0.1563 for Process-Reward scoring. These findings indicate that the evaluation process remains relatively objective and not significantly affected by response length or ordering.

## C  Case Study 1: Why Do Models Perform Poorly?

To gain deeper insights into the challenges of MM-OPERA-Bench tasks, we analyzed the low-scoring answers provided by GPT-4o, Gemini-1.5-Pro, and Gemini-1.5-Flash. This analysis serves a dual

purpose: identifying current limitations of these models and informing future advancements in LVLM design and training methodologies. Specifically, we examined 50 randomly selected low-scoring instances (holistic score $\leq 2$) on both the RIA and ICA tasks for each model, investigating the underlying causes of suboptimal performance. It is noteworthy that, due to the inherent complexity of the tasks, a single response may exhibit multiple limitations, resulting in a cumulative contribution of factors exceeding 100%. Furthermore, we present five illustrative case studies, accompanied by detailed analyses, to facilitate further exploration.

**Perceptual Misalignment (45%).** Models frequently demonstrate an inability to accurately detect salient visual features or to appropriately interpret their significance within the broader associative context. This fundamental perceptual limitation manifests in two primary forms: complete omission of critical visual elements (as exemplified in Case 1, where GPT-4o failed to recognize the QR code embedded within the castle image) or inadequate conceptual abstraction from correctly perceived elements (as illustrated in Case 4, where the model identified visual components but failed to abstract the linguistic concept of "See" from an image depicting an act of looking). These perceptual errors initiate cascading reasoning failures that fundamentally compromise the associative process. More specifically, limitations in image resolution, the presence of visual noise, or a lack of sensitivity to certain visual attributes can lead to perceptual inaccuracies. Furthermore, biases in understanding spatial relationships, relative sizes, and interactions between objects within an image can impede accurate scene interpretation.

**Knowledge Retrieval Gap (48%).** Despite possessing encyclopedic knowledge within their parameters, LVLMs exhibit difficulty in activating relevant information during multimodal association tasks, particularly across cultural, linguistic, and domain boundaries. Case 3 exemplifies this challenge, wherein Gemini-1.5-Flash failed to retrieve cross-cultural knowledge pertaining to "Sanmao," leading to the generation of spurious connections rather than the identification of the genuine linguistic homonym linking Chinese literature and cartoons. Similarly, in Case 5, Gemini-1.5-Pro was unable to access historical knowledge regarding peach baskets as the original basketball hoops, resulting in erroneous pattern identification. This suggests that knowledge activation, rather than mere knowledge possession, represents a significant bottleneck in multimodal associative reasoning. This can be attributed to inefficient knowledge indexing, fragmented knowledge representation, or delayed knowledge updates. Furthermore, inadequate confidence assessment and source attribution mechanisms can hinder the effective utilization of retrieved knowledge.

**Overgeneralization (53%).** When confronted with complex or ambiguous associations, models frequently resort to overly broad and imprecise relationships that lack meaningful specificity. This tendency is clearly demonstrated in Case 1, where GPT-4o defaulted to a generic "creativity" association when unable to identify the more specific "hidden symbols" relationship. Similarly, in Case 4, the model proposed an abstract theme of "emphasis and clarity" rather than recognizing the homophonic relationship between musical notes and their verbal counterparts. This pattern reveals a tendency to prioritize plausibility over clarity when faced with challenging associative tasks. This may be influenced by biases in the training data distribution, favoring frequently occurring association patterns. Furthermore, inaccuracies in assessing association strength and calibrating association confidence can lead to an over-reliance on generalized association patterns.

**Limited Insight and Excessive Caution (23%).** A notable subset of failures stemmed from the models' reluctance to venture beyond superficial observations or to propose connections that require conceptual leaps. Case 2 illustrates this limitation, where Gemini-1.5-Pro correctly identified individual elements (centaur and calendar) but declared them "unrelated" rather than exploring potential symbolic associations through astrological knowledge. This cautious approach restricts the models' ability to discover non-obvious but meaningful connections, a cornerstone of human-like associative thinking. This may be due to pre-programmed constraints that limit the exploration of unconventional reasoning paths. Furthermore, a low tolerance for uncertainty and a high aversion to risk can lead to the adoption of conservative reasoning strategies.

Additionally, we observed a small percentage of cases (approximately 1%) where models declined to engage with certain prompts due to safety or ethical considerations, and technical failures (approximately 2%) where models were unable to properly access all image inputs. These findings collectively underscore the multifaceted challenges inherent in open-ended multimodal association tasks, highlighting the need for advancements in visual perception, cross-domain knowledge activation, and reasoning flexibility to achieve more human-like associative capabilities.

## Case 1 (Remote-Item Association Task)

### Instruction

Describe each image briefly. Analyze and explore the relation between the two images, identifying any possible connections, themes, or shared elements.

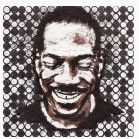 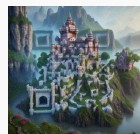

Figure 14: Image 1    Figure 15: Image 2

### Reference Answer

- **Image 1**: A human face made from Oreo cookies
- **Image 2**: A QR code represented by the arrangement of a castle's elements
- **Relation**: Hidden symbols
- **Explanation**: These images use common objects to represent hidden symbols, making Oreo cookies and castle elements serve as symbols for a face and a QR code, respectively.
- **Association Reasoning Path**:
  Symbolize(Oreo, HumanFace) and Is(HumanFace, HiddenSymbols)
  Symbolize(Castle, QRCode) and Is(QRCode, HiddenSymbols)
  Thus, Oreo → HumanFace → HiddenSymbols and Castle → QRCode → HiddenSymbols

### LVLM Output

**GPT-4o:**

- First image: Portrait created with Oreo cookies.
- Second image: Fantasy castle landscape.
- Relation: Creativity.
- Explanation: The first image showcases creativity through an unconventional medium, using Oreo cookies to form a portrait. The second image features a creatively designed fantasy castle, illustrating imaginative architecture and a scenic landscape. Both images highlight artistic expression and inventiveness in their respective forms.

### Judge (GPT-4o) Output

- **Holistic Score**: 2
- **Reason**: The response is somewhat relevant but lacks depth. It correctly identifies the creative aspect of the images but fails to recognize the key theme of 'hidden symbols.' The focus on 'creativity' is too broad and does not align with the intended reasoning.

### Further Analysis

- **Key Words**: Perceptual Misalignment, Overgeneralization
- **Analysis**: GPT-4o's response exhibits both perceptual misalignment and overgeneralization. The model completely misses the QR code hidden within the castle elements (perceptual misalignment), failing to detect the critical visual pattern that would establish the valid association with the Oreo face. This initial perception failure leads to overgeneralization, where the model retreats to a broadly applicable but imprecise "creativity" association rather than identifying the "hidden symbols" that connects both images. This demonstrates how perception failures lead to reasoning limitations, preventing the model from discovering the more sophisticated, intentional symbolic relationships embedded in the visual content.

### Annotation

- **L-3 Perception**: Relational Perception
- **L-3 Conception**: Causal Connections, Thematic Links, Hierarchical Association
- **Relationship Type**: Mutual Element
- **Culture**: N/A
- **Language**: English
- **Topic Domain**: Sense

## Case 2 (Remote-Item Association Task)

### Instruction
Describe each image briefly. Analyze and explore the relation between the two images, identifying any possible connections, themes, or shared elements.

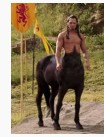

Figure 16: Image 1

Figure 17: Image 2

### Reference Answer
- **Image 1**: The half human and half horse.
- **Image 2**: May and Jun.
- **Relation**: Sagittarius is opposite to Gemini.
- **Explanation**: Sagittarius is the ninth astrological sign in the Western zodiac, represented by the half human and half horse. Gemini is the third astrological sign in the Western zodiac, represented by the twins. Sagittarius and Gemini are opposite signs in the zodiac.
- **Association Reasoning Path**:
  Symbol(HalfHumanHalfHorse, Sagittarius) and Opposite(Sagittarius, Gemini)
  Period(Gemini, MayJun)
  Thus, HalfHumanHalfHorse → Sagittarius → Gemini → MayJun

### LVLM Output
**Gemini-1.5-pro:**
- First image: A centaur, a mythical creature with the upper body of a human and the lower body of a horse.
- Second image: Calendar for the months of May and June 2024.
- Relation: Unrelated
- Explanation: There's no clear connection between the image of a centaur from mythology and a calendar displaying specific months. They originate from entirely different contexts.

### Judge (GPT-4o) Output
- **Holistic Score**: 1
- **Reason**: The LVLM expresses uncertainty and leaves a lack of conceptual connection between the astrological symbols of Sagittarius and the intended zodiac connection.

### Further Analysis
- **Key Words**: Knowledge Retrieval Gap, Limited Insight and Excessive Caution
- **Analysis**: This response demonstrates Limited Insight and Excessive Caution in Gemini-1.5-Pro's associative reasoning. The model fails to activate the deeper astrological knowledge connecting these concepts through zodiac relationships, and retreats to the overly cautious conclusion that the images are "unrelated," reflecting a fundamental unwillingness to venture beyond surface-level observations and preventing the model from making the conceptual leap to recognize that the centaur represents Sagittarius.

### Annotation
- **L-3 Perception**: Relational Perception
- **L-3 Conception**: Causal Connections, Thematic Links, Hierarchical Association
- **Relationship Type**: Relation
- **Culture**: Non-English European
- **Language**: English
- **Topic Domain**: Myth

## Case 3 (Remote-Item Association Task)

### Instruction

Describe each image briefly. Analyze and explore the relation between the two images, identifying any possible connections, themes, or shared elements.

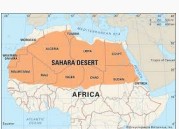 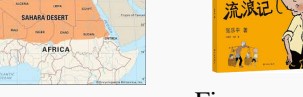

Figure 18: Image 1     Figure 19: Image 2

### Reference Answer

- **Image 1**: Sahara map.
- **Image 2**: *Winter Of Three Hairs.*
- **Relation**: Sanmao (in Chinese).
- **Explanation**: San Mao is a Taiwanese writer who wrote *The Stories of the Sahara.* The story is about her life in the Sahara Desert. The Sahara map is related to the story. Sanmao (Three Hairs) is also a cartoon character created by Zhang Leping in 1935. They share the same Chinese name.
- **Association Reasoning Path**:
  Related(The Stories Of Sahara, Sahara Map) and Write By(SanMao, The Stories Of Sahara)
  SameName(SanMao, Sanmao)
  Thus, SaharaMap → TheStoriesOfSahara → SanMao → Sanmao

### LVLM Output

**Gemini-1.5-flash:**
- First image: Map of the Sahara Desert in Africa
- Second image: Book cover of "Three Hair Travels"
- Relation: A fictional story set in the Sahara Desert
- Explanation: The second image is the book cover of a children's book, "Three Hair Travels," which tells a story in the setting of the Sahara Desert. The first image shows the location and size of the Sahara Desert, which is likely the setting for the story. The book's cover also features imagery that might suggest a desert environment.

### Judge (GPT-4o) Output

- **Holistic Score**: 0
- **Reason**: The LVLM includes incorrect facts, referring to an imaginary story, thus leading to a mistaken chain of reasoning.

### Further Analysis

- **Key Words**: Knowledge Retrieval Gap
- **Analysis**: This response illustrates a significant Knowledge Retrieval Gap in Gemini-1.5-Flash's reasoning process. The model fabricates an entirely fictitious connection by claiming *Winter Of Three Hairs* tells a story set in the Sahara Desert—a complete misrepresentation of this iconic Chinese cartoon about a homeless child's struggles. When directly questioned about *Winter Of Three Hairs* (especially in Chinese), the model fails to activate this knowledge during multimodal association tasks. This disconnect highlights a critical limitation in cross-modal, cross-cultural knowledge retrieval: the model cannot effectively bridge visual perception with cultural knowledge across domains, instead confabulating artificial connections.

### Annotation

- **L-3 Perception**: Semantic Object
- **L-3 Conception**: Cultural Reference
- **Relationship Type**: Relation
- **Culture**: East Asia
- **Language**: Chinese
- **Topic Domain**: Art

## Case 4 (In-Context Association Task)

### Instruction

1. Briefly describe **Image 1**, **Image 2**, and **Image 3** based on their visual information.

2. Analyze the relationship between **Image 1** and **Image 2**, identifying any possible connections, themes, or shared elements that link **Image 1** to **Image 2**.

3. design **Image 4** so that its relationship with **Image 3** mirrors that between **Image 1** and **Image 2**. Use insights from the first pair to guide your design.

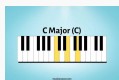 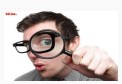 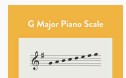

Figure 20: Image 1    Figure 21: Image 2    Figure 22: Image 3

### Reference Answer

- **Image 1**: C Major.
- **Image 2**: See.
- **Image 3**: G Major.
- **Image 4**: Glee.
- **Relation**: Homophones related to musical notes.
- **Explanation**:
  1. The pronunciation of the letter 'C' sounds exactly like the word 'See'.
  2. The pronunciation of 'G' is homophonous with the word 'Glee,' which signifies joy or happiness.
- **Association Reasoning Path**:
  1. Homophone(C, See)
     Thus, C → See
  2. Homophone(G, Glee)
     Thus, G → Glee

## LVLM Output

**GPT-4o:**

- Image 1: A piano keyboard illustration highlighting the C Major chord.
- Image 2: A person holding a magnifying glass up to their eye, creating a humorous enlarged eye effect.
- Image 3: Musical notation for the G Major scale.
- Image 4: A humorous image of a magnifying glass enlarging musical notes or sheet music.
- Relation: Emphasis and Clarity
- Explanation: Image 1 emphasizes specific notes clearly within a piano chord, while Image 2 uses a magnifying glass to humorously emphasize and enlarge a part of a person. Image 4 mirrors this by using a magnifying glass to emphasize specific musical notes from Image 3, establishing a connection through the theme of emphasis and clarity in both musical and visual elements.

---

## Judge (GPT-4o) Output

- **Holistic Score**: 2
- **Reason**: The LVLM identified a loose visual connection (emphasis and clarity) rather than the intended homophone relationship. While the explanation is somewhat reasonable, it does not align with the reference answer's wordplay-based reasoning. The connection between Image 3 and Image 4 is vague and does not capture the intended linguistic homophony.

## Further Analysis

- **Key Words**: Perceptual Misalignment, Overgeneralization
- **Analysis**: The model fails at the perceptual level by describing Image 2 as "a person holding a magnifying glass up to their eye" without abstracting the critical concept of "See," missing its homophonic link to "C Major." This misalignment leads to an overgeneralized interpretation of the pattern as "Emphasis and Clarity." As a result, instead of recognizing "Glee" as the homophonic pair for "G Major," the model suggests "a magnifying glass enlarging musical notes." This illustrates how poor conceptual abstraction undermines pattern recognition, especially in subtle cross-domain associations like linguistic and musical wordplay.

---

## Annotation

- **L-3 Perception**: Abstract Interpretation, Semantic Object
- **L-3 Conception**: Causal Connections, Analogical Reasoning
- **Relationship Type**: Relation
- **Culture**: N/A
- **Language**: English
- **Topic Domain**: Sense, Music

## Case 5 (In-Context Association Task)

### Instruction

1. Briefly describe **Image 1**, **Image 2**, and **Image 3** based on their visual information.

2. Analyze the relationship between **Image 1** and **Image 2**, identifying any possible connections, themes, or shared elements that link **Image 1** to **Image 2**.

3. design **Image 4** so that its relationship with **Image 3** mirrors that between **Image 1** and **Image 2**. Use insights from the first pair to guide your design.

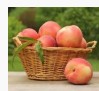
Figure 23: Image 1

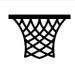
Figure 24: Image 2

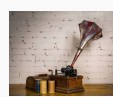
Figure 25: Image 3

### Reference Answer

- **Image 1**: A peach basket.
- **Image 2**: A modern basketball hoop.
- **Image 3**: A phonograph.
- **Image 4**: A modern speaker.
- **Relation**: Evolution of Equipment.
- **Explanation**:
    1. Basketball was originally played with a peach basket as a goal, which evolved into the modern basketball hoop.
    2. The phonograph was an early device for playing recorded sound, which evolved into modern speakers.
- **Association Reasoning Path**:
    1. Evolve(PeachBasket, BasketballHoop)
       Thus, PeachBasket → BasketballHoop
    2. Evolve(Phonograph, Speaker)
       Thus, Phonograph → Speaker

### LVLM Output

**Gemini-1.5-pro:**

- First image: Peaches in a basket.
- Second image: Basketball hoop.
- Third image: Vintage gramophone.
- Forth image: Vinyl records in a case.
- Relation: Container and its contents
- Explanation: Image 1 shows peaches held within a basket, representing a container and its contents. Similarly, Image 2 depicts a basketball hoop, which "contains" a basketball during play. Applying this relationship to Image 3, a gramophone, Image 4 would then be vinyl records, as they are the components "contained" and used by the gramophone to produce sound.

---

### Judge (GPT-4o) Output

- **Holistic Score**: 2
- **Reason**: While the LVLM's reasoning follows a 'container and contents' theme, the connection between a peach in a basket and a basketball in a hoop is weak. A basketball hoop does not function as a container in the same way a basket does. This weakens the overall logical consistency of the response, making it somewhat reasonable but lacking a strong foundation.

### Further Analysis

- **Key Words**: Overgeneralization, Perceptual Misalignment
- **Analysis**: The model fails to retrieve critical knowledge that peach baskets were the original basketball goals. This leads to a "container and contents" relationship, which makes the connection between a peach in a basket and a basketball in a hoop is weak—a basketball hoop does not function as a container in the same way a basket does. This weakens the logical consistency. This illustrates how early perceptual misalignment can cascade, with superficial pattern matching overriding deeper conceptual understanding in multimodal reasoning.

---

### Annotation

- **Relationship Type**: Relation
- **L-3 Perception**: Relational Perception, Semantic Object
- **L-3 Conception**: Causal Reasoning, Functional Links
- **Culture**: N/A
- **Language**: English
- **Topic Domain**: Sports, Music

# D  Case Study 2: Success Cases

A close examination of high-performing instances offers invaluable insights into the model's strengths and the specific characteristics of a high-quality response. These cases demonstrate the model's practical effectiveness and serve as a benchmark for its optimal behavior. We report three such high-scoring cases (HR >= 3).

**Case 1 (Remote-Item Association Task)**

### Instruction

Describe each image briefly. Analyze and explore the relation between the two images, identifying any possible connections, themes, or shared elements.

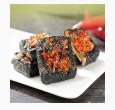

Figure 26: Image 1

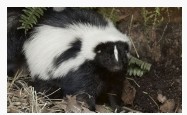

Figure 27: Image 2

### Reference Answer

- **Image 1**: Stinky tofu.
- **Image 2**: A skunk.
- **Relation**: Unpleasant smell.
- **Explanation**: Stinky tofu and skunks are both known for their strong, unpleasant smell.
- **Association Reasoning Path**:
  Smell(StinkyTofu, UnpleasantSmell)
  Smell(Skunk, UnpleasantSmell)
  Thus, StinkyTofu → UnpleasantSmell and Skunk → UnpleasantSmell

### LVLM Output

**GPT-4o:**
- First image: Spicy stinky tofu
- Second image: Skunk in natural habitat
- Relation: Odor
- Explanation: Both images feature subjects known for their strong, distinctive smells. Stinky tofu is a fermented dish famous for its pungent aroma, while skunks are notorious for their powerful defensive spray. Despite their different contexts, both are associated with unique and often polarizing scents.

### Judge (GPT-4o) Output

- **Holistic Score**: 4
- **Reason**: The MLLM correctly identifies the relation as 'odor' and explains it in terms of strong, distinctive smells, consistent with the reference answer's focus on 'unpleasant smell.'

### Further Analysis

- **Analysis**: This case reflects a relatively straightforward instance of associative reasoning. The model correctly identifies each image and retrieves the salient fact that both are known for their distinctive smells, which are frequently emphasized in cultural and zoological contexts. It highlights the model's strength in capturing explicit thematic links from its training data.

### Annotation

- **L-3 Perception**: Semantic Object
- **L-3 Conception**: Thematic Links, Cultural Reference
- **Relationship Type**: Mutual Elements
- **Culture**: Chinese
- **Language**: English
- **Topic Domain**: Sense

**Case 2 (Remote-Item Association Task)**

## Instruction

Describe each image briefly. Analyze and explore the relation between the two images, identifying any possible connections, themes, or shared elements.

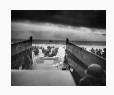

Figure 28: Image 1

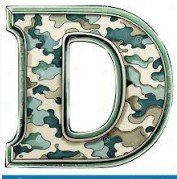

Figure 29: Image 2

## Reference Answer

- **Image 1**: Omaha Beach.
- **Image 2**: Letter D.
- **Relation**: D-Day.
- **Explanation**: Omaha Beach, as before, represents one of the principal landing beaches of the D-Day invasion. The letter D symbolizes the day of the invasion, emphasizing the significance of the operation and the turning point it marked in World War II.
- **Association Reasoning Path**:
  LandingSite(OmahaBeach, DDay)
  Symbol(LetterD, DDay)
  Thus, OmahaBeach → DDay and LetterD → DDay

## LVLM Output

**Gemini-1.5-Pro:**
- First image: Soldiers landing on a beach
- Second image: Letter D in camouflage pattern
- Relation: D-Day
- Explanation: The first image depicts the historical event of D-Day, the Allied invasion of Normandy during World War II. The second image shows the letter "D" in a camouflage pattern, symbolizing the military operation and its association with the letter "D" in "D-Day."

## Judge (GPT-4o) Output

- **Holistic Score**: 4
- **Reason**: The MLLM output is accurate and logically consistent with the reference answer. It correctly identifies the relation as 'D-Day' and provides an explanation that mirrors the standard reasoning by associating the images with the historical event and its symbolism.

## Further Analysis

- **Analysis**: This case requires a more sophisticated form of reasoning, moving beyond shared physical attributes to connect a historical event with its symbolic representation. The model must not only recognize the historical context of the first image (the D-Day landings) but also interpret the second image as a symbolic component of the event's name. The camouflage pattern on the letter 'D' is a crucial cue that reinforces the military theme. This success highlights the model's capacity to bridge concrete visual representations with abstract symbolic knowledge, demonstrating a strong integration of visual analysis with specific historical world knowledge.

## Annotation

- **L-3 Perception**: Contextual Sensory Cues
- **L-3 Conception**: Cultural Reference
- **Relationship Type**: Mutual Elements
- **Culture**: USA/English
- **Language**: English
- **Topic Domain**: History

## Case 3 (In-Context Association Task)

### Instruction

1. Briefly describe **Image 1**, **Image 2**, and **Image 3** based on their visual information.

2. Analyze the relationship between **Image 1** and **Image 2**, identifying any possible connections, themes, or shared elements that link **Image 1** to **Image 2**.

3. design **Image 4** so that its relationship with **Image 3** mirrors that between **Image 1** and **Image 2**. Use insights from the first pair to guide your design.

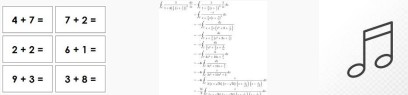

Figure 30: Image 1    Figure 31: Image 2    Figure 32: Image 3

### Reference Answer

- **Image 1**: Basic addition equations.

- **Image 2**: An integral expression.

- **Image 3**: A musical note.

- **Image 4**: A symphony score.

- **Relation**: From Simple to Complex.

- **Explanation**:

  1. Basic addition is a simple, foundational operation, while integration represents a higher-level, complex concept in calculus that builds upon it.

  2. A single note is a basic musical element, while a symphony score is a complex composition made up of many such elements.

- **Association Reasoning Path**:

  1. Addition $\wedge$ Integration $\rightarrow$ From Simple to Complex

  2. MusicalNote $\wedge$ SymphonyScore $\rightarrow$ From Simple to Complex

### LVLM Output

**GPT-4o:**

- First image: Simple math addition problems.

- Second image: A complex mathematical integration problem with detailed steps.

- Third image: A basic musical note symbol.

- Forth image: A complex sheet of musical notes, possibly including a variety of symbols, rhythms, and notations.

- Relation: Complexity progression

- Explanation: The relationship between the first pair of images is based on the complexity of mathematical problems. Image 1 shows basic arithmetic while Image 2 depicts a complex integration problem. Applying this relationship to the second pair, Image 3 shows a simple musical note, so Image 4 should represent a more complex musical composition with a detailed arrangement.

---

### Judge (GPT-4o) Output

- **Holistic Score**: 4

- **Reason**: The MLLM's response correctly identifies the complexity progression relationship between the images and suggests an appropriate complex sheet of musical notes for Image 4. The explanation logically extends the complexity theme from simple math to complex math and from a simple musical note to a complex sheet, aligning well with the reference answer.

### Further Analysis

- **Analysis**: This case showcases a clear instance of cross-domain associative reasoning based on structural progression. The model demonstrates accurate perception, appropriate domain knowledge, and effective abstraction of the "complexity progression" theme. While the reasoning is not particularly challenging due to the explicit visual and semantic cues, it highlights the model's ability to generalize a structural concept across distinct domains.

---

### Annotation

- **Relationship Type**: Relation

- **L-3 Perception**: Abstract Interpretation

- **L-3 Conception**: Hierarchical Association, Analogical Reasoning

- **Culture**: N/A

- **Language**: English

- **Topic Domain**: STEM, Music

## E  Testing and Evaluation Prompts

We report our prompts for testing and evaluation.

### E.1  Testing Prompts

Our testing prompt comprises two principal components. The first component constitutes the instruction for our tasks, and the second delineates the output format for the LVLM. The overall organizational structure is as follows:

Prompt: `<Instruction>` + `<Format>`

The specific implementation of our prompt structure is presented below.

---

**Testing Prompt for RIA Task**

Describe each image briefly.
Analyze and explore the relation between the two images, identifying any possible connections, themes, or shared elements.
Formulate the output as follows:
- First image: [image concept]
- Second image: [image concept]
- Relation: [one keyword, phrase or sentence]
- Explanation: [1-5 sentences]

---

**Testing Prompt for ICA Task**

1. Briefly describe **Image 1**, **Image 2**, and **Image 3** based on their visual information.
2. Analyze the relationship between **Image 1** and **Image 2**, identifying any possible connections, themes, or shared elements that link **Image 1** to **Image 2**.
3. Design **Image 4** so that its relationship with **Image 3** mirrors that between **Image 1** and **Image 2**. Use insights from the first pair to guide your design.
4. Present your analysis in this format:
- **Image 1**: [image concept]
- **Image 2**: [image concept]
- **Image 3**: [image concept]
- **Image 4**: [image concept that you design]
- **Relation**: [a keyword, phrase, or sentence describing the connection]
- **Explanation**: [1-5 sentences detailing the reasoning and its application to both pairs]

---

## E.2 Prompts for Regular LLM-as-a-Judge Scoring

Prompts for Regular LLM-as-a-Judge Scoring comprises three principal components. The first component constitutes the scoring rules, which provide the LLM judge with a five-level scoring gradient (0–4) for evaluating the quality of LVLM responses. The second component delineates the input/output format; in this section, we supply the LLM judge with input references and constrain its output format, thereby enhancing the standardization of the information flow. The third component consists of exemplars. Here, we employ a Few-Shot approach to furnish the LLM judge with concrete examples of scoring criteria, effectively mitigating the high scoring redundancy associated with One-Shot approaches and enhancing scoring diversity, while simultaneously reinforcing the judge's accurate comprehension of the evaluation standards. The overall organizational structure is as follows:

---

Prompt: `<Scoring Rules>` + `<I/O Format>` + `<Rating Examples>`

---

The specific implementation of the prompt structure is presented below.

---

**Scoring Rules for RIA Task**

Evaluate the quality of responses provided by LVLMs using the following criteria:

- **4 points**: Accurate and logically consistent. Convey the same level of thoughtfulness and insight, the same caliber of intellectual rigor and comprehension with reference answer. (Not required to be literally identical to reference answers)

- **3 points**: Shows reasonable understanding but may be incomplete or lack key insights, such as mentioning key facts but failing to effectively summarize the relationship. However, what can be directly got from the images or a common theme with little similarity with the reference answer are not key facts.

- **2 points**: Somewhat relevant but lacks depth, is overly broad, or misses critical elements needed for a coherent explanation; does not address or align with the intended standard reasoning at all.

- **1 point**: Displays uncertainty, offering vague or incomplete reasoning, or concedes with phrases like "I don't know."

- **0 points**: Factuality hallucination in the explanation, i.e. contains factual inaccuracies or fabricated information that undermines its reasoning.

---

### Rating Examples for RIA Task

**Rating examples:**
**Example 1**
LVLM Output:[
- First image: Chicago skyline with Willis Tower and glass buildings.
- Second image: Cloud Gate ("The Bean") sculpture in Millennium Park.
- Relation: Chicago landmarks
- Explanation: Both are iconic Chicago structures - the Willis Tower defining the skyline and Cloud Gate serving as a renowned public sculpture, representing the city's architectural heritage.
]
Reference Answer: [
"relation": "Chicago",
"explanation": "Both Skyscraper and Bean are located in Chicago. Skyscrapers are a prominent feature of Chicago's skyline, and Cloud Gate ("The Bean") is a famous public sculpture located in Millennium Park, Chicago.",
"reasoning": "LocatedIn(Skyscraper, Chicago)
LocatedIn(Bean, Chicago)
Thus, Skyscraper $\rightarrow$ Chicago and Bean $\rightarrow$ Chicago"
]
{
"score_judge": "4",
"score_reason": "The LVLM's output captures the essence of both images as notable Chicago landmarks, consistent with the reference answer, although its summary 'Chicago landmarks' differs from the reference answer 'Chicago.'"
}

*[4 more examples of different scores omitted for brevity]*

---

### Scoring Rules for ICA Task

Background: LVLMs analyze three images, identify relationships between Images 1 and 2, and propose Image 4 that maintains the same relationship with Image 3.
Your task: Evaluate LVLM responses focusing on Image 4 description, Relation, and Explanation against reference answers using these criteria:

- **4 points**: Accurate and logically consistent. Correctly matches reference answer's logic (not necessarily identical). Accurate relation identification and logical Image 4 description.

- **3 points**: Generally correct but incomplete. Shows reasonable understanding but may be incomplete or lack key insights, such as mentioning key facts but failing to effectively summarize the relationship.

- **2 points**: Somewhat reasonable but lacks depth, is overly broad, or misses critical elements needed for a coherent explanation; does not address or align with the intended standard reasoning at all; completely unrelated to the reference answer's focus; completely misses the thematic connection; does not address the intended

theme; a completely different interpretation. Vague relation or loose connection between image 3 and 4.

- **1 point**: Responds with "I don't know," "I can't help," "Unknown connection," etc., or expresses its uncertainty cautiously.

- **0 points**: Contains factual errors or illogical content. Factuality hallucination in the explanation, i.e. contains factual inaccuracies or fabricated information that undermines its reasoning. (Note that hallucination is only related to the LVLM's explanation. That is, even if LVLM's answer does not address or align with the intended standard reasoning at all, it must not get 0 points unless it refers to erroneous information. Divergent opinions on the image theme do not lead to factual hallucination, but wrong image content does.)

---

## Rating Examples for ICA Task

**Rating examples:**
**Example 1**
LVLM Output:[
- Image 1: film Amélie
- Image 2: Eiffel Tower
- Image 3: From Vegas to Macau II
- Image 4: Macau Tower
- Relation: Setting/Location
- Explanation: Image 1 (Amélie) is closely tied to Paris, France, where the film was set and shot. Image 2 shows the Eiffel Tower, a symbol of Paris. Likewise, Image 3 (From Vegas to Macau II) highlights Macau in both its title and visuals. Thus, Image 4 should feature a notable Macau landmark, such as the Macau Tower.
]
Reference Answer: [
"Image 4": "the new Lisboa Hotel", "relation": "Films Associated with Iconic Locations",
"explanation": "Amélie is a film that captures the essence of Paris, showcasing its charm, streets, and culture. Similarly, From Vegas to Macau is centered around the gambling and nightlife culture in Macau, with the New Lisboa Hotel being a significant landmark in that context.",
"reasoning path 1": "FilmSetting(Amelie, Paris)
CulturalSymbol(EiffelTower, Paris)
Thus, Amelie → Paris"
"reasoning path 2": "FilmSetting(MacauStorm, Macau)
CulturalSymbol(NewLisboaHotel, Macau)
Thus, MacauStorm → NewLisboaHotel"
]
{
"score_judge": "4",
"score_reason": "The LVLM accurately linked 'Amélie' with Paris via the Eiffel Tower and 'From Vegas to Macau II' with Macau via the Macau Tower. Although the reference answer highlighted the New Lisboa Hotel for Macau, the Macau Tower is also a valid and recognizable symbol. The LVLM's explanation was clear and logical, with no errors."
}

*[4 more examples of different reasoning omitted for brevity]*

---

## I/O Format for Both RIA and ICA Tasks

You will receive multiple independent questions in a numbered format:

```
1. LVLM Output: []
   Reference Answer: []
```

```
2. LVLM Output: []
   Reference Answer: []
```

Provide your response in JSON format where each key is the question number and the value is your answer:

```
{
    "1": {"score_judge": "", "score_reason": ""},
    "2": {"score_judge": "", "score_reason": ""}
}
```

### E.3 Prompts for LLM Judging in MM-OPERA Reasoning

Our prompt implementation adopts a cross-structured architectural framework and comprises four principal components. The first component establishes the evaluative role, instituting the foundational operational parameters for the LLM judge. The second component formalizes the assessment methodology by constructing a cross-structured prompt that simultaneously provides the LLM judge with both the evaluative task specifications and output format requirements, effectively optimizing the prompt structure and enhancing the consistency of intentional conveyance within the linguistic framework. The third component comprises detailed annotations and format delineations, enabling the LLM judge to integrate task-specific analytical elements while further reinforcing the input-output structural protocol. The fourth component presents calibrated exemplars through a Few-Shot approach to further elucidate the assessment criteria and standardize evaluation procedures. The comprehensive organizational structure of our prompt is as follows:

Prompt: `<Role Definition>` + `<Cross-Structured Instructions>` + `<Annotative Framework and I/O Protocol>` + `<Rating Examples>`

The specific implementation of our prompt structure is presented below. And the sections marked with ellipses share a similar structure and content with the surrounding context and are therefore omitted for brevity.

---

**Role Definition, Cross-Structured Instructions, Annotative Framework and I/O Protocol for RIA Reasoning-guided Evaluation Task**

You are an expert judge evaluating association paths between two image concepts. Your task:
1. Analysis Input

```
input:{
    concepts: [ImageA_desc, ImageB_desc],
    reference_answer: {
    relation: string,
    explanation: string,
    path: string
}
lvlm_output: {
    description: [ImageA_desc, ImageB_desc],
    relation: string,
    explanation: string
    }
}
```

2. Path Standardization for lvlm_output

```
rules:
    {
      type1_sequential: "Predicate_{11}(A, X_{11}) and ...
      and Predicate_{1i}(X_{1i}, X)
      and Predicate_{21}(X, X_{21}) and ...
```

---

```
        and Predicate_{2j}(X_{2j}, B)\n
        A → X_{11} → ... → X_{1i} → X
        → X_{21} → ... → X_{2j} → B"
        type2_convergent: "Predicate_{11}(A, X_{11}) and ...
        and Predicate_{1i}(X_{1i}, X)\n
        Predicate_{21}(B, X_{21}) and ... and Predicate_{2j}
        (X_{2j}, X)\n
        A → X_{11} → ... X_{1i} → X and B → X_{21} → ... → X_{2j} → X"
        type3_metaphorical: "A $\land$ B → X"

        format: {
          - PascalCase for entities/predicates
          - 'and' for clause connection
          - '$\land$' for entity connection
          - '→' for each association hop
        }
    }
```

3. Quality Assessment

```
    for each hop in path:
      score: [
        float[0-1],
        // Reasonableness: logical soundness to the ideal relation X
        float[0-1],
        // Distinctiveness: The clarity of concept boundaries
        int[0 or 1]
        // Knowledgeability: Domain knowledge depth
      ]
```

Note: The path don't need to be identical to the reference answer but should demonstrate equivalent quality of reasoning.
You will receive multiple independent problem-answer pairs in a numbered format:
1. Problem: ...
2. Problem: ...
Provide your response in JSON format where each key is the problem-answer pair number and the value is your evaluation:

```
{"1": {...},"2": {...}}
```

## Rating Examples for RIA Reasoning-guided Evaluation Task

**Examples 1:**
Problem:
- First image: Bread and Wine
- Second image: 12
Reference Answer:
- Relation: The Last Supper
- Explanation: The Last Supper is a religious painting depicting the final meal Jesus shared with his disciples before his crucifixion. The bread and wine represent the elements of the Eucharist, a central sacrament in Christianity. The number 12 refers to the twelve apostles present at the Last Supper.
- Association Path: Depict(TheLastSupper, Eucharist) and Represent(BreadAndWine, Eucharist) Number(TheLastSupper, 12)
Thus, BreadAndWine → Eucharist → TheLastSupper and 12 → TheLastSupper
LVLM's Output:
- First image: A loaf of bread and a chalice
- Second image: The number 12 in gold
- Relation: Christian symbolism

- Explanation: The first image is a visual representation of the Eucharist, a central sacrament in Christianity, where bread and wine are consecrated and consumed as the body and blood of Jesus Christ. The number 12 holds significant symbolism in Christianity, representing the 12 apostles of Jesus. Together, these images allude to a key aspect of Christian faith and tradition.

Your evaluation:

{
"path": "Represent(BreadAndWine, Eucharist) and Symbolize(Eucharist, Christian)
Number(12, ApostlesOfJesus) and Related(ApostlesOfJesus, Christian)
Thus, BreadAndWine $\rightarrow$ Eucharist $\rightarrow$ Christian and 12 $\rightarrow$ ApostlesOfJesus $\rightarrow$ Christian",
"hop_quality": {
"BreadAndWine $\rightarrow$ Eucharist": [1.00, 1.00, 1],
"Eucharist $\rightarrow$ Christian": [1.00, 1.00, 1],
"12 $\rightarrow$ ApostlesOfJesus": [1.00, 1.00, 1],
"ApostlesOfJesus $\rightarrow$ Christian": [1.00, 1.00, 1]
}
"explanation": "The LVLM constructed a well-structured convergent path through Christian symbolism. Both paths (BreadAndWine$\rightarrow$Eucharist$\rightarrow$Christian and 12$\rightarrow$ApostlesOfJesus$\rightarrow$Christian) demonstrate perfect scores across all dimensions, showing deep theological understanding and precise use of religious concepts."
}

*[2 more examples of different scores omitted for brevity]*

## Role Definition, Cross-Structured Instructions, Annotative Framework and I/O Protocol for ICA Reasoning-guided Evaluation Task

LVLMs analyze three images, identify relationships between Images 1 and 2 (Pair 1), and propose Image 4 that maintains the same relationship with Image 3 (Pair 2).
You are an expert judge evaluating LVLM's association path. Your task:
1. Analysis Input

```
input: {
 concepts: [Image1_desc, Image2_desc, Image3_desc],
 reference_answer: {
   image4: Image4_desc
   relation: string,
   explanation: string,
   path: string
 },
 lvlm_output: {
   description: [Image1_desc, Image2_desc, Image3_desc, Image4_desc],
   relation: string,
   explanation: string
 }
}
```

2. Path Standardization for lvlm_output

```
   rules:
   {
     type1_sequential: "Predicate_{11}(A, X_{11}) and ...
     and Predicate_{1i}(X_{1i}, X) and Predicate_{21}(X, X_{21})
     and ...
     and Predicate_{2j}(X_{2j}, B)\n
     A → X_{11} → ... → X_{1i} → X → X_{21} → ... → X_{2j} → B"
     type2_convergent: "Predicate_{11}(A, X_{11}) and ...
     and Predicate_{1i}(X_{1i}, X)\n
     Predicate_{21}(B, X_{21}) and ...
```

```
        and Predicate_{2j}(X_{2j}, X)\n
        A → X_{11} → ... X_{1i} → X and B → X_{21} → ... → X_{2j} → X"
        type3_metaphorical: "A $\land$ B → X"

        format: {
          - PascalCase for entities/predicates
          - 'and' for clause connection
          - '$\land$' for entity connection
          - '→' for each association hop
        }
    }
```

3. Quality Assessment for lvlm_output

```
    for each hop in path:
      score: [
        float[0-1],
        // Reasonableness: logical soundness to the ideal relation X
        float[0-1],
        // Distinctiveness: The clarity of concept boundaries
        int[0 or 1]
        // Knowledgeability: Domain knowledge depth
      ]
```

Note: The path don't need to be identical to the reference answer but should demonstrate equivalent quality of reasoning. If the lvlm_output is irrelevant or incorrect, provide a low score for each hop.
You will receive multiple independent problem-answer pairs in a numbered format:
1. Problem: ...
2. Problem: ...
Provide your response in JSON format where each key is the problem-answer pair number and the value is your evaluation:

```
{"1": {...},"2": {...}}
```

---

**Rating Examples for ICA Reasoning-guided Evaluation Task**

**Examples 1:**
Problem:
- Image 1: Burj Khalifa
- Image 2: Canton Tower
- Image 3: Statue of Liberty
Reference Answer:
- Image 4: Eiffel Tower
- Relation: Built in the Same Year
- Explanation 1: The Burj Khalifa in Dubai and Canton Tower in Guangzhou were both completed in 2010, marking a significant year for architectural achievements.
- Association Path 1: BuiltIn(BurjKhalifa,2010)
BuiltIn(CantonTower,2010)
Thus, BurjKhalifa → 2010 and CantonTower → 2010
- Explanation 2: The Statue of Liberty and the Eiffel Tower were both inaugurated in 1886, symbolizing important cultural landmarks in their respective countries.
- Association Path 2: BuiltIn(StatueOfLiberty,1886)
BuiltIn( EiffelTower, 2010)
Thus, StatueOfLiberty → 1886 and EiffelTower → 1886
LVLM's Output:
- **Image 1**: The Burj Khalifa, the tallest building in the world, located in Dubai, United Arab Emirates.
- **Image 2**: Canton Tower, also the tallest tower in China, located in Guangzhou,

Guangdong Province, China.
- **Image 3**: The Statue of Liberty, a colossal neoclassical sculpture on Liberty Island in New York Harbor, New York City.
- **Image 4**: The Eiffel Tower, a wrought iron lattice tower on the Champ de Mars in Paris, France.
- **Relation**: "Tallest Structures"
- **Explanation**: All four images depict some of the tallest structures in the world, each representing a different architectural style and cultural significance. Image 1 and Image 2 are skyscrapers, while Image 3 is a statue, and Image 4 is an iconic tower. This common theme allows for a comparison of how these structures have become symbols of their respective cities and countries.
Your evaluation:
{
"path1": "Is(BurjKhalifa, Tallest)
Is(CantonTower, Tallest)
BurjKhalifa → Tallest and CantonTower → Tallest", "path2": "Is(StatueOfLiberty, Tallest)
Is(EiffelTower, Tallest)
StatueOfLiberty → Tallest and EiffelTower → Tallest",
"hop_quality_path1": {
"BurjKhalifa → Tallest": [0.95, 0.86, 1],
"CantonTower → Tallest": [0.95, 0.85, 1]
},
"hop_quality_path2": {
"StatueOfLiberty → Tallest": [0.55, 0.45, 1],
"EiffelTower → Tallest": [0.83, 0.85, 1]
},
"explanation": "In the LVLM's output, the first path shows consistently high hop quality scores as both the Burj Khalifa and Canton Tower are indeed among the tallest structures. The second path shows more varied scores, with the Statue of Liberty receiving lower scores as it's not typically categorized among the world's tallest structures, while the Eiffel Tower maintains high scores."
}

*[1 more examples of different reasoning omitted for brevity]*

# F   Limitations and Broader Impacts

## F.1   Limitation

While MM-OPERA represents a significant advancement in evaluating association reasoning in Large Vision-Language Models, several limitations highlight areas for future refinement.

- **Limited Exploration of Temporal Association Reasoning**: MM-OPERA's static task design (RIA and ICA) does not fully capture temporal or sequential association reasoning, a key aspect of human cognition in dynamic contexts like decision-making, restricting its evaluation scope.

- **High Cost and Scalability Challenges for Open-Ended Evaluation**: Evaluating 11,497 open-ended tasks with a resource-intensive LLM-as-a-Judge and cascading scoring rubric incurs high computational costs (due to increased token usage) and limits scalability, hindering rapid or large-scale testing of LVLMs.

- **Challenges in Systematic Task Creation**: Although association is common in human cognition, systematically collecting and converting ideas or existing data into task instances is challenging, especially given LLMs' weaknesses in this area, leading to high human effort costs for data expansion.

These limitations underscore the need for continued innovation to enhance MM-OPERA's robustness, scalability, and applicability in advancing AI research.

## F.2 Broader Impacts

MM-OPERA is an **evaluation** benchmark for associative reasoning in Large Vision-Language Models (LVLMs), not a training set. It aims to deepen understanding and guide AI development. However, evaluation standards carry societal implications.

**Potential Societal Considerations:**

- **Guiding Development and Bias Risks:** Benchmarks shape research. Any unaddressed gaps or subtle biases within MM-OPERA could inadvertently steer development towards a narrow or skewed form of associative intelligence, impacting real-world fairness and applicability.

- **Perception of Capabilities and Misuse Potential:** By identifying models with advanced associative abilities, MM-OPERA may elevate perceptions of their power. Such identified capacities, even if not developed via this benchmark, could be leveraged for sophisticated misuse (e.g., disinformation) if not responsibly managed.

- **Deployment Risks from Identified Limitations:** MM-OPERA reveals model weaknesses. Overlooking these identified limitations during deployment in critical systems could lead to erroneous and harmful outcomes.

**Mitigation and Responsible Use of Insights:** Transparency in design and responsible interpretation of MM-OPERA's results are crucial. We advocate for:

- Continuous community scrutiny and refinement of MM-OPERA to address potential biases and representational gaps.

- Using evaluation insights to understand fundamental AI limitations and guide research towards robust, safe, and aligned systems, beyond mere model ranking.

- Informed deployment decisions by developers and deployers, using benchmarked strengths and weaknesses to assess suitability and mitigate risks.

Our goal is for MM-OPERA to foster rigorous evaluation, contributing to more capable and societally beneficial AI.

