# OpenReview forum: "MM-OPERA: Benchmarking Open-ended Association Reasoning for Large Vision-Language Models"
_NeurIPS.cc/2025/Datasets_and_Benchmarks_Track — NeurIPS 2025 Datasets and Benchmarks Track poster_

### Official Review · Reviewer_9E1P · 2025-06-29

**Rating:** 5
**Confidence:** 4

**Summary:**

This paper constructs the MM-OPERA benchmark dataset to evaluate LVLMs' association reasoning capabilities. Additionally, the authors design various evaluation criteria, using LLMs to score LVLMs' association reasoning. Finally, the paper conducts extensive experiments on various LLMs, including sensitivity analysis of task instances, validity analysis of LLM-as-a-Judge strategies, and diversity analysis across abilities, domains, languages, cultures, etc. These studies provide a comprehensive and nuanced understanding of the limitations of current language models in terms of associative reasoning.

**Dataset Code Accessibility:**

Yes

**Dataset Code Comments:**

1.The dataset tables and code are clearly accessible.

**Ethical Considerations:**

No, there are no or only very minor ethics concerns

**Limitations Weaknesses:**

1.The "human baseline" mentioned in section 5.1 consists of only 24 students, which likely does not adequately represent human performance, neither in terms of quantity nor scope of the survey. It is recommended to expand the survey scope or modify the metric name accordingly.

2.It is suggested to add some success cases in the appendix (for example, cases where HR >= 3).

**Strengths Contributions:**

1.The paper is generally well-written and easy to follow. The figures, tables, and captions are clear and easy to understand.

2.The paper makes significant contributions to the problem of LVLMs' association reasoning, including the construction of the MM-OPERA benchmark dataset and LLM-based evaluation criteria, paving the way for more human-like and general-purpose AI.

3.For the two tasks, RIA and ICA, the authors conduct extensive experimental analyses to comprehensively evaluate LVLMs' association reasoning capabilities.

4.In the appendix, the data construction, multi-dimensional experiment analysis, and generated examples are clearly and reliably explained. The dataset and code are clearly presented and transparent, making the results reproducible.

---

> ### Author Rebuttal · Authors · 2025-07-28
>
> Thank you for your kind recognition of our work and your constructive suggestions.
>
> ---
>
> **Question 1**: The "human baseline" mentioned in section 5.1 consists of only 24 students, which likely does not adequately represent human performance, neither in terms of quantity nor scope of the survey. It is recommended to expand the survey scope or modify the metric name accordingly.
>
> **Answer 1**: Thanks for prompting this important improvement. We agree that a larger and more diverse sample would better reflect human performance and appreciate the reviewer’s suggestion.
>
> Our current baseline, while limited in scale, was intentionally composed of cognitively strong undergraduate and graduate students with diverse majors. Due to the demanding nature of open-ended tasks, we prioritized quality over quantity in the initial phase.
>
> We recognize the need for broader coverage. We are planning a larger-scale study, including a more demographically diverse pool. This will yield a more robust statistical baseline and allow exploration of group-level performance differences.
>
> ---
>
> **Question 2**: It is suggested to add some success cases in the appendix.
>
> **Answer 2**: Thank you for this valuable suggestion. We will add a section to the appendix in the camera-ready version with several high-scoring cases (HR >= 3). Since external pages and images are not allowed in the rebuttal, here we provide brief descriptions of two cases we plan to include.
>
> ---
> **Case 1 (Remote-Item Association Task)**
>
> **Reference Answer**:
>
> - **Image 1**: Stinky tofu
> - **Image 2**: A skunk
> - **Relation**: Unpleasant smell.
> - **Explanation**: Stinky tofu and skunks are both known for their strong, unpleasant smell.
> - **Association Reasoning Path**:
>
>   Smell(StinkyTofu, UnpleasantSmell)
>
>   Smell(Skunk, UnpleasantSmell)
>
>   Thus, StinkyTofu → UnpleasantSmell and Skunk → UnpleasantSmell
>
> **LVLM (GPT-4o) Output**:
>
> - **First image**: Spicy stinky tofu
> - **Second image**: Skunk in natural habitat
> - **Relation**: Odor
> - **Explanation**: Both images feature subjects known for their strong, distinctive smells. Stinky tofu is a fermented dish famous for its pungent aroma, while skunks are notorious for their powerful defensive spray. Despite their different contexts, both are associated with unique and often polarizing scents.
>
> **Judge Output**:
>
> - **Holistic Score**: 4
> - **Reason**: The MLLM correctly identifies the relation as 'odor' and explains it in terms of strong, distinctive smells, consistent with the reference answer's focus on 'unpleasant smell.'
>
> **Further Analysis**: This case reflects a relatively straightforward instance of associative reasoning. The model correctly identifies each image and retrieves the salient fact that both are known for their distinctive smells, which are frequently emphasized in cultural and zoological contexts. It highlights the model's strength in capturing explicit thematic links from its training data.
>
> **Annotation**:
>
> - **L-3 Perception**: Semantic Object
> - **L-3 Conception**: Thematic Links, Cultural Reference
> - **Relationship Type**: Mutual Elements
> - **Culture**: Chinese
> - **Language**: English
> - **Topic Domain**: Sense
>
>
> ---
> **Case 2 (In-Context Association Task)**
>
> **Reference Answer**:
>
> - **Image 1**: Basic addition equations
> - **Image 2**: An integral expression
> - **Image 3**: A musical note
> - **Image 4**: A symphony score
> - **Relation**: From Simple to Complex
> - **Explanation**:
>   1. Basic addition is a simple, foundational operation, while integration represents a higher-level, complex concept in calculus that builds upon it.
>   2. A single note is a basic musical element, while a symphony score is a complex composition made up of many such elements.
> - **Association Reasoning Path**:
>   1. Addition ∧ Integration  → From Simple to Complex
>   2. MusicalNote ∧ SymphonyScore → From Simple to Complex
>
> **LVLM (GPT-4o) Output**:
>
> - **Image 1**: Simple math addition problems.
> - **Image 2**: A complex mathematical integration problem with detailed steps.
> - **Image 3**: A basic musical note symbol.
> - **Image 4**: A complex sheet of musical notes, possibly including a variety of symbols, rhythms, and notations.
> - **Relation**: Complexity progression
> - **Explanation**: The relationship between the first pair of images is based on the complexity of mathematical problems. Image 1 shows basic arithmetic while Image 2 depicts a complex integration problem. Applying this relationship to the second pair, Image 3 shows a simple musical note, so Image 4 should represent a more complex musical composition with a detailed arrangement.
>
> **Judge Output**:
>
> - **Holistic Score**: 4
> - **Reason**: The MLLM's response correctly identifies the complexity progression relationship between the images and suggests an appropriate complex sheet of musical notes for Image 4. The explanation logically extends the complexity theme from simple math to complex math and from a simple musical note to a complex sheet, aligning well with the reference answer.
>
> **Further Analysis**: This case showcases a clear instance of cross-domain associative reasoning based on structural progression. The model demonstrates accurate perception, appropriate domain knowledge, and effective abstraction of the "complexity progression" theme. While the reasoning is not particularly challenging due to the explicit visual and semantic cues, it highlights the model’s ability to generalize a structural concept across distinct domains.
>
> **Annotation**:
>
> - **L-3 Perception**: Abstract Interpretation
> - **L-3 Conception**: Hierarchical Association, Analogical Reasoning
> - **Relationship Type**: Relation
> - **Culture**: N/A
> - **Language**: English
> - **Topic Domain**: STEM, Music
>
> ---
>
> We hope that our responses have sufficiently addressed your concerns, and we welcome any further feedback you may have.

---

> > ### Comment · Reviewer_9E1P · 2025-08-07
> > **Comments by Reviewer 9E1P**
> >
> > Thank you for your response. I will keep the previous score.

---

> ### Comment · Area_Chair_9sqB · 2025-08-06
>
> Dear Reviewer,
>
> Please engage with the authors' rebuttal and provide your final recommendation (clear Accept/Reject, avoiding borderline scores) by August 8 AoE. Submit your "Mandatory Acknowledgement" only after completing these steps. Your prompt attention to this is greatly appreciated.
>
> Best regards,
>
> Area Chair, NeurIPS DB 2025

---

> ### Author Response · Authors · 2025-08-08
>
> Dear Reviewer 9E1P,
>
> We sincerely thank you for your kind response and for recognizing our efforts to address your concerns. We will ensure that the revisions and clarifications discussed during the rebuttal are reflected in the paper.
>
> Best regards,
>
> Authors of Paper 1120

---

### Official Review · Reviewer_MHP7 · 2025-07-01

**Rating:** 5
**Confidence:** 5

**Summary:**

The paper investigates the intelligence capability of humans, association, by expanding the closed-ended tasks to open-ended tasks. They propose RIA and ICA to challenge the model’s divergent thinking and convergent associative thinking. Additionally, they provide a comprehensive analysis of the limitations of current MLLMs.

**Dataset Code Accessibility:**

Yes

**Ethical Considerations:**

No, there are no or only very minor ethics concerns

**Limitations Weaknesses:**

1. The formulation should be improved, such as the long-tail sentence.
2. How can the future work use association to design next-generation MLLMs?
3. How do you ensure the quality of data with true association? How do you ensure the quality of LLM as a Judge?
4. What is the true reason that MLLMs lie behind humans?
5. What are the potential ways to use association in MLLMs pre-training? Is there an association-related loss that needs to be optimized to improve association ability? Or what potential ways can improve association ability?
6. Is it possible to evaluate The Labyrinth of Links on proposed benchmarks?
7. How to combine the conclusion in The Labyrinth of Links with your benchmark to give a solid analysis for current MLLMs. And provide some insights for how to utilize these benchmarks or how to improve performance on athe ssociation task?

**Strengths Contributions:**

1. The paper investigates the important and underexplored capability in human association, expanding the deficiency of existing work.
2. The paper proposes the RIA and ICA tasks that support the open-ended evaluation.

---

> ### Author Rebuttal · Authors · 2025-07-28
>
> Thank you for your kind recognition of our work and your constructive feedback.
>
> ---
>
> **Question 1**: The formulation should be improved, such as the long-tail sentence.
>
> **Answer 1**: We agree on the importance of clarity. We will simplify complex sentences to improve readability and ensure our contributions are communicated clearly.
>
> ---
>
> **Question 2**: How can the future work use association to design next-generation MLLMs?
>
> **Answer 2**: MM-OPERA and its underlying principles offer several directions for advancing LVLMs:
>
> - **Association-Rich Training Data**: Our work highlights a "creativity gap" (ΔHR metric). MM-OPERA's design offers a blueprint for creating open-ended, diverse instruction-tuning datasets that promote divergent thinking.
> - **Improved Knowledge Activation**: We reveal a "retrieval gap": models often hold relevant knowledge but fail to connect it across domains. This motivates research into associative memory architectures that retrieve concepts via abstract links, enabling multi-hop reasoning.
> - **Targeted Skill Enhancement**: MM-OPERA’s taxonomy of 13 associative abilities enables fine-grained diagnosis. This can guide targeted training and modular designs where varied components specialize in specific reasoning types.
> - **Scalable Process-Supervised Fine-Tuning**: Our Process-Reward Judge evaluates reasoning processes. Future work can extend this to training by generating and scoring reasoning paths at scale, enabling process-supervised fine-tuning that improves coherence and creativity in association.
>
> ---
>
> **Question 3**: How do you ensure the quality of data with true association, and the quality of LLM-as-a-Judge?
>
> **Answer 3**:
>
> **1) Ensuring Data Quality with True Associations**: We ensure the validity of every task instance through a rigorous, multi-stage process detailed in Appendix A.4.
>
> - Two-step Human Review: a **Cross-Review** by multiple annotators for correctness and clarity, followed by an **Expert Review** by core researchers to confirm the conceptual soundness and depth of the association.
> - **Structured, Traceable Association Reasoning Path** allowed our reviewers to scrutinize the step-by-step logical chain. It provides a robust and objective measure of correctness.
>
> **2) Ensuring the Quality and Reliability of the LLM-as-a-Judge:**
>
> - Our **advanced, transparent framework of Process-Reward LLM-as-a-Judge** (Sec. 4.2) moves beyond simple outcome-based scoring to provide a fine-grained assessment of the entire reasoning process. This gives us a more transparent and nuanced understanding of a model's capabilities.
> - We rigorously tested our judge's reliability (Sec. 5.5, Appendix B.4) and confirmed:
>   - **Negligible Bias**: We found only minimal verbosity bias (Pearson’s r < 0.38) and negligible position bias (mean score difference < 0.16 across permutations).
>   - **Strong Human Alignment**: Our judge's scores show remarkable alignment with human experts, achieving 78.33% perfect matches in regular scoring. For our PR-Judge, we measured strong correlations with domain-expert scores for Reasonableness (r=0.72) and Distinctiveness (r=0.68), and high accuracy for Knowledgeability (83.5%, Kappa=0.65).
> - **Mitigating Self-Enhancement Bias**: To further reduce bias, we employ a dual-judge approach using both GPT-4o and DeepSeek-V3, a powerful open-source model from a different developer with distinct architecture. This ensures an independent evaluation perspective.
>
> ---
>
> **Question 4**: What is the true reason that MLLMs lie behind humans?
>
> **Answer 4**: While a "true reason" is elusive, our case study in Appendix C, points to some core limitations behind the gap in associative reasoning (one response may exhibit multiple limitations). MM-OPERA expose the following gaps, guiding development toward more human-like associative AI.
>
> - **Weak Conceptual Grounding**: Unlike humans who learn through multisensory interaction, LVLMs rely on statistical correlations, leading to perceptual misalignment (45% of failures). They may fail to recognize objects or miss their contextual implication.
> - **Brittle Knowledge Activation**: Humans fluidly connect distant ideas, but LVLMs exhibit a knowledge retrieval gap (48% of failures). Their knowledge is vast but inert, often dependent on surface-level co-occurrence. They struggle to link concepts across domains or modalities unless explicitly trained to do so.
> - **Optimized for Plausibility, Not Insight**: Current training objectives (e.g., next-token prediction, RLHF) favor safe, plausible responses over insightful ones, leading to:
>   - **Overgeneralization** (53%): Retreating to a vague concept when unsure, which lacks specific insight.
>   - **Excessive Caution** (23%): Safety alignment often penalizes speculation, leading models to be conservative rather than attempting a creative leap.
>
> ---
>
> **Question 5**: What are the potential ways to use association in MLLMs pre-training? Is there an association-related loss that needs to be optimized to improve association ability? Or what potential ways can improve association ability?
>
> **Answer 5**: While our work focuses on evaluation, our findings suggest several promising for integrating association into MLLM pre-training:
>
> - **Contrastive Association Objective**: Reinforcement learning is increasingly part of the MLLM training/alignment phase (e.g., RLHF, RLVF). This opens the door for novel, association-centric objectives. We can formulate a contrastive loss, treating conceptually-linked multimodal pairs as positive samples and unlinked or random pairs as negative samples. This directly teaches the model to identify and encode deep, non-obvious relationships.
> - **Association Path Prediction**: Inspired by our reasoning-path evaluation, pre-train models can predict missing nodes in conceptual chains mined from large corpora. This encourages multi-hop reasoning and addresses the knowledge retrieval gap.
> - **Divergent Generative Pre-training**: To reduce overgeneralization and cautiousness, pre-training can reward models for generating diverse, plausible associations. This requires novel objectives that go beyond next-token prediction, possibly drawing from reinforcement learning to capture both quality and diversity.
>
> In sum, enhancing association ability likely requires new objectives that explicitly model relational, sequential, and divergent thinking. Our benchmark provides the tools to evaluate such progress.
>
> ---
>
> **Question 6**: Is it possible to evaluate The Labyrinth of Links (LoL) on proposed benchmarks?
>
> **Answer 6**: LoL and MM-OPERA operate under fundamentally different paradigms. As summarized in Table 1:
>
> - LoL evaluates association across adjectives and verb semantic concepts in a closed-ended setting using multiple-choice tasks, emphasizing scalability and step-by-step retrieval along a predefined path.
> - MM-OPERA targets open-ended associations, requiring models to generate novel connections and coherent reasoning paths.
>
> Though it is inappropriate to evaluate LoL on MM-OPERA, the two benchmarks are complementary. Your question suggests a promising future direction: integrating LoL’s dynamic structure into MM-OPERA’s generative setting. Future work can design multi-turn, open-ended tasks where each generated association becomes the seed for the next, blending both paradigms into a dynamic, generative reasoning chain.
>
> ---
>
> **Question 7**: How to combine the conclusion in LoL with your benchmark to give a solid analysis for current MLLMs. And provide some insights on how to utilize these benchmarks or how to improve performance on the association task?
>
> **Answer 7**: The two benchmarks reveal complementary deficiencies:
>
> - LoL exposes weaknesses in structured, sequential association. Its closed-ended format highlights models' inability to maintain coherent associative chains over time, pointing to instability in memory and retrieval.
> - MM-OPERA reveals deficits in generative, creative association, requiring models to form and justify novel connections. Failures like the Creativity Gap and Knowledge Retrieval Gap indicate poor conceptual flexibility and reasoning.
>
> A shared underlying cause may be the lack of training on unpaired data, as LoL suggests. It explains MM-OPERA’s failure patterns: without exposure to conceptually related but infrequent pairs, models default to overgeneralization or caution when faced with novel associations.
>
> **How to utilize the benchmarks**:
>
> - **As diagnostic tool for model development**: The fine-grained evaluation (e.g., Association Reasoning Path, failure modes like Knowledge Retrieval Gap and Perceptual Misalignment) helps identify why models fail at complex reasoning, informing architectural and training improvements toward more human-like associative intelligence.
> - **As a prerequisite capability benchmark for downstream applications**. Associative reasoning is essential for domains such as
>   - **Scientific Discovery**: Generating novel hypotheses by connecting disparate findings.
>   - **Creative Ideation**: Brainstorming non-obvious concepts for design, marketing, or humor.
>   - **Embodied AI & Robotics**: Repurpose objects for new functions, indicating generalization potential in unstructured settings.
>   - **Personalized Education**: Generate insightful analogies for effective tutoring
>
> **How to improve model performance**: The benchmark results offer valuable insights for improving model performance through better pre-training objectives, fine-tuning strategies, and data collection practices. These findings can inform the development of more effective alignment and reasoning mechanisms. We elaborate on these directions in **Answer 2** and **Answer 5**.
>
> ---
>
> We hope our responses resolve your questions.

---

> > ### Comment · Reviewer_MHP7 · 2025-08-05
> >
> > Thanks for your detailed response. Hope the part of the discussion contents will appear in the final version. I keep my positive score.

---

> ### Author Response · Authors · 2025-08-08
>
> Dear Reviewer MHP7,
>
> We sincerely thank you for your kind response and for recognizing our efforts to address your concerns. We will ensure that the revisions and clarifications discussed during the rebuttal are reflected in the paper.
>
> Best regards,
>
> Authors of Paper 1120

---

### Official Review · Reviewer_i3RA · 2025-07-03

**Rating:** 4
**Confidence:** 2

**Summary:**

The paper introduces MM-OPERA, a benchmark designed to evaluate open-ended association reasoning in Large Vision-Language Models (LVLMs). Unlike traditional benchmarks that rely on closed-ended tasks, MM-OPERA assesses LVLMs' ability to perform creative and complex associative reasoning, inspired by human cognitive processes like convergent and divergent thinking.

**Dataset Code Accessibility:**

No

**Dataset Code Comments:**

The authors do not provide the codes or datasets in the paper.

**Ethical Considerations:**

No, there are no or only very minor ethics concerns

**Limitations Weaknesses:**

1. Temporal Reasoning: Does not assess dynamic or sequential associations.
2. Scalability: High computational costs for open-ended evaluation.
3. Data Curation: Manual effort required for task creation.
4. Other advanced models: Qwen-2.5-VL[1], Kimi-VL[2], and MiniMax-VL-01[3] are not included in the main experiments. I sugget the authors conduct more experiments on these advanced models.
5. About Link-Context Ability: although MM-OPERA focus on ICL ability of MLLMs, it is better to also incorporate LCL[4] ability to further make the benchmark more faithful. Beyond similar demonstrations, there are also indirect relationships that can highly correspond to LCA scenarios.

[1] Bai S, Chen K, Liu X, et al. Qwen2. 5-vl technical report[J]. arXiv preprint arXiv:2502.13923, 2025.
[2] Team K, Du A, Yin B, et al. Kimi-vl technical report[J]. arXiv preprint arXiv:2504.07491, 2025.
[3] Li A, Gong B, Yang B, et al. Minimax-01: Scaling foundation models with lightning attention[J]. arXiv preprint arXiv:2501.08313, 2025.
[4] Tai Y, Fan W, Zhang Z, et al. Link-context learning for multimodal llms[C]//Proceedings of the IEEE/CVF Conference on Computer Vision and Pattern Recognition. 2024: 27176-27185.

**Strengths Contributions:**

MM-OPERA Benchmark Comprises 11,497 instances across two tasks:
1. Remote-Item Association (RIA): Linking seemingly unrelated concepts (e.g., connecting an armadillo and Kevlar fabric via their protective functions).
2. In-Context Association (ICA): Recognizing and transferring associative patterns (e.g., linking a bald eagle to basketball via their American origins, then applying this to a lion and football).

It spans 13 associative dimensions, diverse cultures, languages, and domains to ensure comprehensive evaluation.

---

> ### Author Rebuttal · Authors · 2025-07-28
>
> Thank you for your recognition of our work and your constructive feedback.
>
> ---
>
> We believe **Questions 1–3** are best addressed by first clarifying a core distinction between MM-OPERA and prior closed-ended benchmarks like *The Labyrinth of Links* [1]. While [1] made foundational contributions, its advantages like scalability, automated data creation, and potential for temporal reasoning, are intrinsically tied to its core limitation: closed-ended tasks with multiple-choice format. This format, by its nature, cannot capture the open-ended, generative, and creative essence of true associative reasoning.
>
> MM-OPERA addresses this gap by focusing on open-ended association, a more difficult yet meaningful challenge. The concerns raised reflect thoughtful trade-offs we made in pursuit of this goal, and as outlined below, we’ve taken concrete steps to mitigate them.
>
> **Question 1**: Temporal Reasoning: Does not assess dynamic or sequential associations.
>
> **Answer 1**: We intentionally avoid dynamic, interactive feedback, which are effective only in closed, predefined settings but is far from open-ended association. It would artificially constrain the vast search space of creative association.
>
> Instead, MM-OPERA **supports sequential reasoning implicitly aligned with the Association Reasoning Path**, encouraging models to form concept bridges across multiple hops. **The Process-Reward LLM-as-a-Judge** requires models to internally construct and express multi-hop concept chains, thus tests deeper cognitive capabilities beyond selecting from predefined options.
>
> **Question 2**: Scalability: High computational costs for open-ended evaluation.
>
> **Answer 2**: The use of LLM-as-a-Judge with inevitable high computational costs is a deliberate and necessary choice. Open-ended associative reasoning demands semantic nuance. Rigid, rule-based evaluation would penalize novel and creative responses that deviate from a fixed "correct" answer, thereby defeating the entire purpose of the benchmark.
>
> To reduce costs and enhance scalability, we support local deployment of our LLM-as-a-Judge strategies using advanced open-source model (e.g., DeepSeek-V3). Thanks to the community’s contributions, it significantly reduces reliance on proprietary APIs and improves long-term scalability and reproducibility.
>
> **Question 3**: Data Curation: Manual effort required for task creation.
>
> **Answer 3**: We aim to build challenging and meaningful association tasks that probe deep knowledge. The manual effort invested in MM-OPERA is essential for its quality and difficulty, since automated generation risks triviality and shallowness.  To scale efficiently, we used a **seed-and-expand strategy** (Appendix A.4): core researchers created exemplars, trained volunteers built upon them, and concepts from validated psychometric tests (RAT, LI-RAT) were adapted to LVLMs. Our multi-stage quality control ensures robustness and consistency.
>
> In sum, MM-OPERA reflects a deliberate shift toward evaluating deeper, open-ended intelligence. We believe the insights gained are well worth the trade-offs.
>
> [1] Li, Hong, et al. "The Labyrinth of Links: Navigating the Associative Maze of Multi-modal LLMs." The Thirteenth International Conference on Learning Representations.
>
> ---
>
> **Question 4**: Other advanced models: Qwen-2.5-VL, Kimi-VL, and MiniMax-VL-01 are not included in the main experiments. I suggest the authors conduct more experiments on these advanced models.
>
> **Answer 4**: Thank you for suggesting these recent models. Regarding MiniMax-VL-01, we were unfortunately unable to include it during the rebuttal period due to its significant size (456B params) and the current lack of a publicly accessible API. For the other two models, the evaluation results are as follows:
>
> **Remote-Item Association (RIA) Task**:
>
> |Model|SR(%)|HR-4(%)|HR-3(%)|$\Delta$HR(%)|
> |---|:---:|:---:|:---:|:---:|
> |Qwen2.5-VL-7B-Instruct|52.28|5.35|20.36|15.00|
> |Kimi-VL-A3B-Instruct|48.41|5.14|16.43|11.30|
>
> **In-Context Association (ICA) Task**:
>
> |Model|SR(%)|HR-4(%)|HR-3(%)|$\Delta$HR(%)|
> |---|:---:|:---:|:---:|:---:|
> |Qwen2.5-VL-7B-Instruct|53.50|1.08|16.62|15.54|
> |Kimi-VL-A3B-Instruct|48.96|0.94|14.17|13.22|
>
> The results reinforce the primary conclusions of our paper.
>
> 1. Qwen-2.5-VL-7B shows a strong performance, positioning it at the top of the open-source models we tested. Kimi-VL also ranking competitively at the third of the open-source models we tested. However, a clear gap remains when compared to leading proprietary models (e.g., Gemini-2.5-Pro's 60.05% and 63.09% SR) and our human reference (61.88% and 68.69% SR), confirming the limitations of current LVLMs in associative reasoning.
> 2. The significant gap between their overall score rates (SR) and their ability to generate high-quality, insightful associations (very low HR-4 scores, especially on the ICA task) aligns with our paper's finding: while current models can often produce reasonable connections, generating truly creative insights remains a challenge.
> 3. A key observation lies in the $\Delta$HR metric, which proxies for divergent thinking (generating reasonable but non-optimal associations). Notably, **Qwen-2.5-VL exhibits one of the highest $\Delta$HR scores among all tested open-source models** on both tasks. This suggests that while it may not always produce the single most insightful answer (low HR-4), it possesses a strong capability for creative flexibility, generating a wider breadth of plausible associative connections.
>
> We will add the results to the main experiment tables to our paper. We are grateful for the suggestion, as it further strengthens the comprehensiveness and timeliness of our study.
>
> ---
>
> **Question 5**: About Link-Context Ability: although MM-OPERA focus on ICL ability of MLLMs, it is better to also incorporate LCL [2] ability to further make the benchmark more faithful. Beyond similar demonstrations, there are also indirect relationships that can highly correspond to LCA scenarios.
>
> **Answer 5**: We agree that **In-Context Learning (ICL)** and **Link-Context Learning (LCL)** represent fundamentally different paradigms. ICL focuses on **pattern recognition** from task-irrelevant demonstrations, while LCL emphasizes **causal relevance**, requiring the model to extract and apply specific rules or mappings from the support set to answer the query. In LCL, MLLMs are expected to assimilate new concepts from ongoing context and retain this knowledge for accurate reasoning. For example, [2] introduces the ISEKAI dataset, where models must learn AI-generated novel concepts and distinguish them from real-world counterparts.
>
> Integrating a meaningful LCL track into MM-OPERA is non-trivial. To remain consistent with our benchmark’s emphasis on complex, open-ended reasoning, LCL tasks would need to go beyond simple concept mappings. They should require models to **infer abstract, novel rules** from demonstrations and apply them through **multi-step associative reasoning**. Designing such tasks and constructing sufficient convincing cases demand substantial human-LVLM cooperation, which is difficult to accomplish within the limited timeframe of the rebuttal period.
>
> Nevertheless, we recognize the great value of this direction and will add it to our discussion section. We plan to develop a dedicated LCL track in future iterations of MM-OPERA. Thank you for this insightful suggestion.
>
> [2] Tai, Yan, et al. "Link-context learning for multimodal llms." Proceedings of the IEEE/CVF Conference on Computer Vision and Pattern Recognition. 2024.
>
> ---
>
> We hope our responses have addressed your concerns, and we welcome any further feedback you may have.

---

> ### Comment · Area_Chair_9sqB · 2025-08-06
>
> Dear Reviewer,
>
> Please engage with the authors' rebuttal and provide your final recommendation (clear Accept/Reject, avoiding borderline scores) by August 8 AoE. Submit your "Mandatory Acknowledgement" only after completing these steps. Your prompt attention to this is greatly appreciated.
>
> Best regards,
>
> Area Chair, NeurIPS DB 2025

---

> ### Author Response · Authors · 2025-08-08
>
> Dear Reviewer i3RA,
>
> I hope this message finds your well. We sincerely appreciate the time and effort you have devoted to providing thoughtful and constructive feedback. As the Author-Reviewer discussion phase is nearing its ends with **less than 12 hours remaining**, we want to ensure we have addressed all your concerns satisfactorily. Please let us know if you have any additional feedback. We genuinely hope that the improvements and clarifications we have made will be considered in your final assessment.
>
> Thank you again for your valuable feedback.
>
> Best regards,
>
> Authors of Paper 1120

---

### Official Review · Reviewer_pnjR · 2025-07-03

**Rating:** 5
**Confidence:** 2

**Summary:**

This paper introduces a new benchmark and dataset for evaluating open-ended associative reasoning. The authors consider Remote-Item Association (RIA), which models must find a conceptual link between two unrelated images or concepts, and In-Context Association (ICA), of which models are asked to extrapolate association patterns from a context pair to a new query.

**Additional Feedback:**

* How do the authors think of balancing the number of samples across different languages?

**Dataset Code Accessibility:**

Yes

**Dataset Code Comments:**

The provided datasets are accessible well

**Ethical Comments:**

I found no ethics concerns in the proposed benchmark

**Ethical Considerations:**

No, there are no or only very minor ethics concerns

**Final Justification:**

This paper introduceㄴ a benchmark with various strengths including exploration on overlooked aspects and diverse case of samples. While I initially had some concerns about the benchmark. the authors resolved them with detailed responses. Thus, I'll maintain my positive rating.

**Limitations Weaknesses:**

* The proposed benchmark is limited to synthetic data. It's unclear how performance here transfers to practical, downstream tasks like robot planning or education.
* Although using multiple models like GPT-4o and DeepSeek helps reduce bias, relying on LLMs to evaluate other LLMs may still at risks.

**Strengths Contributions:**

* This paper address overlooked aspects about 'open-ended associative reasoning'
* Analyzed several MLLMs and human baselines
* Diverse datasets
    * The proposed datasets contain 11,497 curated instances, with multilingual and cross-cultural coverage (15 languages, 22 domains). Its tasks also vary in difficulty (e.g., 0–5 reasoning hops), fostering robust benchmarking.

---

> ### Author Rebuttal · Authors · 2025-07-28
>
> Thank you for your kind recognition of our work and your insightful questions.
>
> ---
>
> **Question 1**: The proposed benchmark is limited to synthetic data. It's unclear how performance here transfers to practical, downstream tasks like robot planning or education.
>
> **Answer 1**: We agree that the ultimate goal of AI research is to impact real-world applications. Associative reasoning is a foundational cognitive skill. Although our benchmark uses controlled tasks rather than simulating a full downstream application, it is precisely this control that allows us to isolate and reliably measure this fundamental ability. We posit that strong associative ability is a prerequisite for success in many practical domains. For instance:
>
> - **Robot Planning**: A robot in a novel environment cannot rely on pre-programmed instructions. It must improvise. Imagine a robot needs to retrieve an object from a high shelf and sees a sturdy box and a long pole nearby. A simple model might see two unrelated items. A model with strong associative ability, like that tested in our Remote-Item Association (RIA) task, could form the non-obvious connection: "box as a stable base" and "pole as a reaching tool," creating a novel solution. This is not just object recognition; it is a creative, goal-directed synthesis of function.
> - **Education**: An advanced AI tutor's effectiveness hinges on its ability to create powerful analogies. If a student is struggling with the abstract concept of "economic inflation," a model skilled in In-Context Association (ICA) could abstract the pattern "more input (money) leads to less value" and apply it to a context the student understands, such as "in a video game, when everyone has a legendary sword, it's no longer special or valuable." This requires abstracting a relational principle and reapplying it creatively.
>
> Although explicitly quantifying this performance transfer is a vital avenue for future work, our benchmark provides the first essential tool to enable such studies. By offering a robust method to measure associative intelligence, MM-OPERA can serve as a diagnostic tool for researchers in robotics, education, and beyond, helping them understand why their models fail at complex tasks and inspiring the development of more human-like reasoning architectures.
>
> We will clarify this perspective in our revision, framing our contribution as the critical first step of isolating and measuring a foundational capability, thereby enabling future research on its direct application.
>
> ---
>
> **Question 2**: Although using multiple models like GPT-4o and DeepSeek helps reduce bias, relying on LLMs to evaluate other LLMs may still be at risk.
>
> **Answer 2**: We are fully aware of the potential risks associated with LLM-as-a-Judge and have made its validation a central part of our contribution.
>
> First, we want to emphasize that our choice of LLM-as-a-Judge was a principled decision driven by the **open-ended** nature of the task. Since practical associative reasoning is often open-ended, demanding divergent thinking and multiple creative solutions, current closed-ended benchmarks cannot measure this vital capability. A rule-based judge with fixed "correct" answers would unfairly penalize novel, creative, and even superior associations that diverge from a predefined reference. Such a rigid approach would stifle the very creativity we aim to measure. Therefore, a nuanced judge capable of understanding semantics is necessary.
>
> Having made this choice, we implemented a comprehensive strategy to mitigate risks and ensure our evaluation is as robust and unbiased as possible:
>
> - **A more transparent evaluation framework**: We moved beyond standard outcome-based scoring. Our novel **Process-Reward LLM-as-a-Judge** (PR-Judge) provides a fine-grained assessment of the entire reasoning chain (Sec 4.2). This method offers superior transparency and nuance by evaluating individual reasoning steps, allowing us to distinguish between elegant and flawed reasoning paths even when they reach similar conclusions.
> - **Extensive empirical validation and human alignment**: We rigorously tested our judge's reliability against common biases and human judgment (Sec. 5.5 and Appendix B.4). Our key findings confirm the judge's robustness:
>   - **Negligible bias**: We found only minimal verbosity bias (Pearson’s r < 0.38) and negligible position bias (mean score difference < 0.16 across permutations).
>   - **Strong human alignment**: Our judge's scores show remarkable alignment with human evaluators. For regular scoring, we found 78.33% perfect matches with no discrepancies exceeding 1 point. For our PR-Judge, we measured strong correlations with domain-expert scores for Reasonableness (r=0.72) and Distinctiveness (r=0.68), and high accuracy for Knowledgeability (83.5%, Kappa=0.65).
>
> In conclusion, although no LLM-based evaluation is entirely without risk, we have proactively addressed the challenge by designing a more transparent evaluation process and demonstrating its strong alignment with human experts and low susceptibility to common biases. We believe this rigorous validation makes our findings reliable and our framework a robust tool for the community.
>
> ---
>
> **Question 3**: What do the authors think of balancing the number of samples across different languages?
>
> **Answer 3**: Thanks for this insightful question. The current distribution (in Fig. 2e) is a principled design choice to robustly evaluate distinct aspects of associative reasoning.
>
> - **English serves as the dominant language**: Current LVLMs are primarily trained on English-centric data, so using English as the primary language offers a fair and stable baseline for evaluating core capabilities.
> - To achieve a more comprehensive evaluation, we additionally include **non-English samples**, which serve as targeted probes for **culturally nuanced associative phenomena**. These include challenges such as **linguistic wordplay** (e.g., homophones or puns unique to a given language) and **cultural knowledge** (e.g., proverbs, historical references, or artistic expressions that require deep, language-specific world knowledge).
> - **Why non-English samples follow a long-tail distribution**: The design of non-English samples requires the cultural context associated with each language, resulting in more complex construction logic and thus a more limited pool of suitable examples.
>
> To provide a more balanced view of multilingual capability, we now report the **harmonic mean** ($H_{SR}$) of the score rate (SR) on English and non-English samples, and compute  of the two. This metric mitigates the dominance of the larger English subset and is sensitive to performance disparities across languages. The following tables will be included in the revised paper.
>
> **Remote-Item Association (RIA) Task**:
>
> |Model|$H_{SR}$ (%)|Eng SR (%)|Non-English SR (%)|
> |---|:---:|:---:|:---:|
> |**Proprietary LVLMs**||||
> |Claude-3.5-Sonnet|52.92|48.37|58.41|
> |Gemini-1.5-Flash|58.12|55.28|61.27|
> |Gemini-1.5-Pro|49.70|44.17|56.80|
> |Qwen-VL-Max|51.05|44.84|59.27|
> |Qwen-VL-Plus|47.47|41.24|55.90|
> |Gemini-2.0-Flash-Thinking-Exp|60.93|58.58|63.47|
> |Gemini-2.5-Pro-Preview|**63.53**|58.96|68.87|
> |o4-mini|62.39|59.74|65.28|
> |GPT-4o|62.02|59.27|65.04|
> |**OpenSource LVLMs**||||
> |GLM-4V|32.48|25.00|46.35|
> |InternVL-Chat-V1-2|40.72|35.14|48.41|
> |InternLM-XComposer2.5-7B|**51.36**|49.98|52.82|
> |VILA1.5|49.10|46.14|52.45|
> |Yi-VL-34B|50.26|43.98|58.65|
>
> **In-Context Association (ICA) Task**:
>
> |Model|$H_{SR}$ (%)|Eng SR (%)|Non-English SR (%)|
> |---|:---:|:---:|:---:|
> |**Proprietary LVLMs**||||
> |Claude-3.5-Sonnet|48.94|49.66|48.25|
> |Gemini-1.5-Flash|50.13|51.72|48.64|
> |Gemini-1.5-Pro|41.63|42.55|40.74|
> |Qwen-VL-Max|46.22|51.50|41.92|
> |Qwen-VL-Plus|42.93|46.05|40.21|
> |Gemini-2.0-Flash-Thinking-Exp|59.56|62.72|56.71|
> |Gemini-2.5-Pro-Preview|**61.05**|64.53|57.94|
> |o4-mini|59.90|62.72|57.32|
> |GPT-4o|56.94|59.21|54.83|
> |**OpenSource LVLMs**||||
> |GLM-4V|42.68|44.32|41.15|
> |InternVL-Chat-V1-2|31.08|36.37|27.14|
> |InternLM-XComposer2.5-7B|43.52|45.83|41.44|
> |VILA1.5|42.17|46.00|38.92|
> |Yi-VL-34B|**53.51**|55.04|52.06|
>
> - The ​$H_{SR}$ confirms that top models like Gemini-2.5-Pro and o4-mini demonstrate strong, balanced capabilities. This reinforces our conclusions while offering a more nuanced view of multilingual performance.
> - We observe an interesting fact that in the **RIA task**, most models achieve a higher SR on non-English samples. We hypothesize this is because these instances often test specific, well-defined cultural knowledge (e.g., proverbs). A successful association relies on retrieving a precise factual link, which models with broad world knowledge can do effectively. Conversely, in the **ICA task**, many top models perform better on English samples. ICA demands abstracting and transferring a relational pattern, a meta-reasoning skill. We posit this capability is more robustly developed for English, the primary language in pre-training data, where such abstract logical structures are more prevalent.
>
> Note that although some items were designed with specific cultural or cultural contexts in mind, models may still generate alternative but valid associations without explicitly relying on those cues. Our **open-ended** evaluation rewards any well-justified reasoning path, so scores may reflect reasoning flexibility or general knowledge rather than direct cultural or linguistic awareness. Thus, performance differences cannot be solely attributed to language proficiency.
>
> We will include the harmonic mean results in our paper to reflect both absolute performance and cross-lingual robustness more accurately.
>
> ---
>
> We hope our responses have addressed your concerns, and we welcome any further feedback you may have.

---

> > ### Comment · Reviewer_pnjR · 2025-08-04
> >
> > I would like to thank the authors for their detailed responses. I have read all of them and found that my concerns have been well addressed. Thus, I'll maintain my positive rating.

---

> ### Author Response · Authors · 2025-08-08
>
> Dear Reviewer pnjR,
>
> We sincerely thank you for your kind response and for recognizing our efforts to address your concerns. We will ensure that the revisions and clarifications discussed during the rebuttal are reflected in the paper.
>
> Best regards,
>
> Authors of Paper 1120

---

### Official Review · Reviewer_rHto · 2025-07-03

**Rating:** 4
**Confidence:** 4

**Summary:**

This paper introduce MM-OPERA, a large scale multimodal evaluation dataset focus on the evaluation of open-ended association reasoning. MM-OPERA consists of two tasks, Remote-Item Association (RIA) and In-Context Association (ICA), which are designed to be aligned with human psychometric principles.

MM-OPERA is motivated by human's Remote Associates Test (RAT), a human psychological test accesing human's associative ability. MM-OPERA is built on an enhanced setting where each question require the model to generated an open-ended answer with detailedly chain of thought reaoning.

MM-OPERA contains 11497 data instances, spanning diverse modalities concepts and reasoning complexities. Due to its open-ended natrue, MM-OPERA is evaluated by LLM-as-a Judge with detailed step-wise scores.

By experiment, the authors show that LVLM's ability on assiciation reasoning fall behind human, together with more detailed analysis on LVLM's limitaions in such domain.

**Dataset Code Accessibility:**

Yes

**Ethical Considerations:**

No, there are no or only very minor ethics concerns

**Final Justification:**

The authors response addressed most of my concerns, I raised my score to 4.

One remianing concern is still about visual dependency, while some examples indeed need detailed and accurate visual perception, most images are just used to represent one name/concept. This is evidenced by Figure1, where it is sufficient to just recogonize the object name in the image to answer all the question, and this is very common based on my observation from the hf dataset.

**Limitations Weaknesses:**

1. Vision modality is not always necessary in this dataset. From the examples on page2, both questions can be decomposed into captioning + text-based reasoning. The captioning step is not a challenge to current LVLMs, this means MM-OPERA is primarily evaluating LVLM's abiliy of text-based reasoning.

2. The motivation is not convincing enough. In 3.1, the authors only mentioned the RAT exam for humans and that MM-OPERA is more complex than RAT. The reason why current LVLMs need such an evaluation is not mentioned, for example, is it closely related to an important LVLM's application? The reviewer can not come up with any real-world application of the ability that MM-OPERA is evaluaing. As a result, it is suggested to significantly elaborate this aspect.

3. From Table2, human's performance on many tasks are far from perfect (e.g. 22.84%). This brings two possibilities, a. the correctness of the dataset or the evaluation setting is flawed. b. the dataset is too challenging. It is also crucial to clearify this part, and if the dataset is too challenging, the authors should find another way to justfy the overall correctness of the dataset.

**Strengths Contributions:**

1. The dataset is carefully curated, focused, providing a useful testbed for LVLM development.
2. The ayalysis is detailed, diagnostic, and the reliability is well-justified.

---

> ### Author Rebuttal · Authors · 2025-07-28
>
> Thanks for all your invaluable feedback and suggestions.
>
> ---
>
> **Question 1**: Vision modality seems not always necessary in this dataset. Questions can be decomposed into captioning + text-based reasoning. The captioning step is not a challenge to current LVLMs. This means MM-OPERA primarily evaluates LVLM's ability of text-based reasoning.
>
> **Answer 1**: Although the tasks can be decomposed into captioning and text reasoning in principle, our findings reveal that this view oversimplifies the process and the visual modality is actually indispensable and revealing.
>
> - **The visual modality offers an information channel for association that language cannot fully substitute**. Many critical cues, such as style and ambiance, are apprehended visually first, and language serves as a subsequent and often imperfect descriptor for the visual cues. The rich nuances lost in textual abstraction are often the basis for the most insightful associative reasoning.
> - **Captioning is easy, but capturing all visual information is not easy for current LVLMs**. For example, in Case 1 of our case study (Appendix C, p27) GPT-4o captioned the second image as "a fantasy castle," failing to recognize that the castle's elements are arranged to form a QR code. This perceptual failure makes the intended "hidden symbols" association impossible. Similarly, [1] found that LVLMs can struggle to extract visual information, such as reading an analog clock and counting fingers on various hands.
> - **We empirically prove that modality directly impacts reasoning**. In our Text-Image Substitution Test (Sec. 5.4, Appendix B.3.2), replacing one image with its text description (Img-Txt) caused GPT-4o's performance to drop on 49.8% of samples, with 16.8% suffering a severe ≥2-point decline. This suggests vision is more effective for GPT-4o on this task. In contrast, Gemini-1.5-Pro only scored lower on 13.7% of samples under Img-Txt condition. When replacing both images with text descriptions, Gemini-1.5-Pro scored higher on 45.5% of samples and worse on only 9.5%, indicating the visual modality may introduce complexity and ambiguity for it. MM-OPERA can help understand how models leverage different modalities for complex cognitive tasks. Removing vision would obscure these critical insights.
>
> [1] Roberts, Jonathan, et al. "Zerobench: An impossible visual benchmark for contemporary large multimodal models." arXiv preprint arXiv:2502.09696 (2025).
>
> ---
>
> **Question 2**: The motivation seems not convincing enough. The reason why current LVLMs need such an evaluation is not mentioned, for example, is it closely related to an important LVLM application? It is suggested to significantly elaborate this aspect.
>
> **Answer 2**: We are grateful for the opportunity to make our motivation clearer. In cognitive science, associative ability predicts higher-order skills like inference, creativity, and detecting subtle connections. We argue that LVLMs need to develop this core capability to move beyond shallow pattern matching toward true knowledge synthesis and reasoning. It is a prerequisite for many real-world applications:
>
> - **Scientific Discovery**: Generating novel hypotheses requires connecting disparate findings across different papers, domains, or disciplines.
> - **Creative Ideation and Design**: Brainstorming non-obvious concepts for design, marketing, or even generating culturally relevant humor by bridging different semantic domains.
> - **Embodied AI & Creative Problem-Solving**: Enabling an agent to perform novel tool use by associating objects with their functional affordances in new contexts (e.g., recognizing a box can serve as a platform or a stick as a reaching tool).
> - **Personalized Education**: Crafting effective analogies by associating a student's known interests (e.g., video games) with new, complex topics (e.g., physics principles).
>
> Since practical associative reasoning is often open-ended, demanding divergent thinking and multiple creative solutions, current closed-ended benchmarks cannot measure such vital capability. MM-OPERA aims to fill this gap. By challenging models to form and articulate non-obvious links, we directly assess this core competency required for more helpful, insightful, and general-purpose AI. We will add this sharpened motivation and the possible applications to our revised paper.
>
> ---
>
> **Question 3**: Human's performance on many tasks are far from perfect, bringing two possibilities: a. the correctness of the dataset or the evaluation setting is flawed; b. the dataset is too challenging. If b, another way to justify the overall correctness of the dataset should be found.
>
> **Answer 3**: We argue this gap reflects the genuine difficulty of the task (b), not flaws in the dataset or evaluation (a).
>
> - **Our process ensures validity**. We took extensive measures to ensure the quality of MM-OPERA. As detailed in Appendix A.4.2 (Quality Control), every instance underwent a rigorous multi-stage validation process involving: (1) **Cross-Review**: Initial checking by multiple, uninvolved annotators to ensure correctness. (2) **Expert Review**: Final verification by core researchers to confirm conceptual soundness, remove ambiguities, and check for biases.
> - **Another way to justify the overall correctness**: Crucially, what sets our validation apart is the **structured Association Reasoning Path** included with every instance thanks to our Process-Reward LLM-as-a-Judge method. Reviewers validated the entire step-by-step logical chain, ensuring that the association is not only plausible but also coherently and correctly reasoned. This **traceable reasoning** provides a far more robust and objective measure of correctness than benchmarks with only a final label, significantly mitigating the risk of flawed or ambiguous examples.
> - **The difficulty of our benchmark is by design and is informative**. The fact that human performance is far from perfect is consistent with decades of psychometric research like [2] (**the average human performance on the Remote Associates Test was 34.2%** and it was rather low.) The performance gap between human and LVLMs is therefore an important scientific finding about the current state of AI.
>
> In summary, our rigorous, path-based quality control ensures correctness, and the benchmark's deliberate difficulty is essential for its purpose: to reliably measure the significant gap that still exists between current LVLMs and robust associative reasoning. We will revise the main paper to clarify this point.
>
> [2] Ansburg, P. I., & Hill, K. (2003). Creative and analytic thinkers differ in their use of attentional resources. Personality and Individual Differences, 34(7), 1141-1152.
>
> ---
>
> We hope that our responses have addressed your concerns, and we welcome any further feedback you may have.

---

> > ### Comment · Reviewer_rHto · 2025-08-06
> >
> > Thanks for the rebuttals. Here are my responses:
> > 1. The visual modality offers an information channel for association that language cannot fully substitute. Captioning is easy, but capturing all visual information is not easy for current LVLMs.
> >
> > I fully agree that visual modality is an additional information channel that language cannot fully substitute. But based on the examples from the paper and from the huggingface page, it is sufficient to answer the question just with a few words of descriptions of the images in most of them. I also agree that fine-grained visual perception is still challenging for LVLMs, but I can not see any visual challenge presented in this benchmark as in Zero-bench. In addition, the experiment in Appendix B.3.2 only shows that modality impacts reasoning, which I agree, but it doesn't show the how visually challenging the benchmark is, which is my main concern here. To justify this, it is recommended to design a visual perception subtask, to ask the model the specific property that we want them to exatract from the image, which can give people better understanding on model's main bottleneck of this benchmark.
> >
> > 2. Motivation.
> >
> > Thanks for the response of the motivation. I agree that the assiciation reasoning ability is relevant to the ability of scientific discovery, and creativity. But such corrlation seems to be indirect since the examples in MM-OPERA are all real-world objects while scientific discovery often need the combination of new abstract concepts. It is possible that the evaluation conflate the model's knowldedge with model's creativity. Embodied-AI requires precise low-level visual perception and physics understanding, which is not included in the current benchmark.
> >
> > 3. Correctness
> >
> > Thanks for the clearification, it would be great to briefly mention this in the main paper as well.

---

> > ### Comment · Area_Chair_9sqB · 2025-08-08
> >
> > Dear Reviewer rHto,
> >
> > Thank you for your thorough review. As we finalize decisions, I'd like to highlight key points from the discussion for your consideration:
> >
> > 1. The authors demonstrated visual modality's impact through empirical results (49.8% performance drop when replacing images with text)
> >
> > 2. They've strengthened motivation by connecting associative reasoning to higher-order AI applications
> >
> > 3. Five other reviewers supported this benchmark's value
> >
> > Please submit your final Accept/Reject recommendation **(avoiding borderline scores)**. If maintaining rejection, please ensure your justification aligns with core NeurIPS criteria.
> >
> > Your perspective remains valuable to this decision.
> >
> > Best regards,
> >
> > Area Chair, NeurIPS DB 2025

---

> > > ### Author Response · Authors · 2025-08-08
> > >
> > > Dear AC, NeurIPS DB 2025:
> > >
> > > We sincerely thank for your effort to interpret our responses for Reviewer rHto. We have also provided some new responses and aim to address all the concerns by Reviewer rHto.
> > >
> > > Best regards,
> > >
> > > Authors of Paper 1120

---

> ### Author Response · Authors · 2025-08-06
>
> Dear Reviewer rHto,
>
> We sincerely appreciate the time and effort you have devoted to providing thoughtful and constructive feedback. We hope our responses and clarifications have adequately addressed your concerns. As the Author-Reviewer discussion phase draws to a close, please feel free to let us know if you have any further questions or suggestions. We genuinely hope that the improvements and clarifications we have made will be considered in your final assessment.
>
> Thank you again for your valuable feedback.
>
> Best regards,
>
> Authors of Paper 1120

---

> ### Comment · Area_Chair_9sqB · 2025-08-06
>
> Dear Reviewer,
>
> Please engage with the authors' rebuttal and provide your final recommendation (clear Accept/Reject, avoiding borderline scores) by August 8 AoE. Submit your "Mandatory Acknowledgement" only after completing these steps. Your prompt attention to this is greatly appreciated.
>
> Best regards,
>
> Area Chair, NeurIPS DB 2025

---

> ### Author Response · Authors · 2025-08-08
>
> We sincerely thank for Reviewer rHto’s constructive feedback and we also thank for AC to further interpret our responses for Reviewer rHto. We attempt to address Reviewer rHto’s remaining concerns as below.
>
> ## About The Concerns in Visual Challenge (Part 1)
>
> **Q1**: “It is sufficient to answer the question just with a few words of descriptions of the images in most of them."
>
> **A1**: Reviewer rHto misunderstood the **visual challenge** in LVLM benchmark, which **is not equivalent with the number of words in the image’s description**. For instance, identifying a "QR code" within a castle or the exact time "4:33" on a clock (both of them are the instances in our benchmark) only require a few of words but particularly difficult for all evaluated baselines. These cases are common in our benchmark.
>
> **Q2**: "I can not see any visual challenge presented in this benchmark as in Zero-bench. "
>
> **A2:  Reviewer rHto’s judgment “can not see any visual challenge” is arbitrary and unjustified.**
>
> First, our benchmark aims to comprehensively access associative reasoning of MLLM rather than curating adversarial red-teaming cases to promise the visual challenge e.g. ZeroBench. If every LVLM is blind to the visual content, evaluating its associative reasoning ability becomes meaningless (we are able to achieve this by red-teaming like ZeroBench yet it does not align with our motivation).
>
> Second, in case of Reviewer rHto "can not see any visual challenge", we enrich the failure analysis in our benchmark. With regards to manual verification, across GPT-4o, Gemini-1.5-Pro & Flash, **45% of errors stem from Perceptual Misalignment**, primarily due to:
> - **Missed cues** (e.g., Case 1: overlooked QR code in a castle)
> - **Poor abstraction** (e.g., Case 4: seeing “a person...” but missing the concept “See”)
> - **Dataset Composition** (Fig 2): **53.57%** of samples are in the "Perception" ability category (L-1), covering 7 visual sub-dimensions, which are designed to require non-trivial visual understanding.
>
> Our next version will also include diverse reasoning instances with significant visual challenges, some of which are shown below:
>
> - **4'33'' (Detail Recognition)**
>     - Img 1: a clock at 4:33
>     - Img 2: a pianist
>     - Both models misread the time (4o saw '7:25', Pro saw '4:35'). However, when given the correct time as text, both correctly identified the link to John Cage's silent composition 4'33''.
>
>  - **911 (Object Identification)**
>     - Img 1: Porsche 911
>     - Img 2: September 11 attacks
>     - Pro identified both and the common number "911." But 4o only recognized a "luxury car" for Img 1, missing the specific model "911" for the precise association.
>
>   - **The Bluest Eye (Person Recognition)**
>     - Img 1: author Toni Morrison
>     - Img 2: a blue eye
>     - 4o saw a generic "portrait of a person," leading to a weak association. Pro correctly identified "author Toni Morrison," allowing it to connect to her novel, *The Bluest Eye*.
>
>   - **1997 (Event Recognition)**
>     - Img 1: Hong Kong handover ceremony
>     - Img 2: George Soros analyzing the Asian financial crisis
>     - Pro successfully identified both events and found both events occurred in 1997. 4o described Img 2 as "Man holding a photograph," leading to a vague "Historical events" association.
>
>   - **Hermes (Object/Figure Identification)**
>     - Img 1: a lyre
>     - Img 2: marble statue of Hermes
>     - Pro identified both and linked them via "In Greek mythology, the god Hermes is credited with inventing the lyre." 4o described Img 2 as an "Ancient statue of a man," missing "Hermes," thus making a general "Greek mythology" association.
>
> -  **8 (Counting & Cultural Context)**
>     - Img 1: Chinese painting of the Eight Immortals
>     - Img 2: French film still "8 Women" (group of eight women)
>     - Pro correctly identified both and their shared count of "8 people." 4o merely described "mythical figures" and "group of women," failing to count or recognize the cultural context crucial for "8".
>
> The cases, even for top-tier LVLMs like GPT-4o (4o) and Gemini-2.5-Pro-Preview (Pro), still fail due to their visual challenges behind.
>
> **Q3**: "The experiment in Appendix B.3.2 only shows that modality impacts reasoning, but it doesn't show the how visually challenging the benchmark is, which is my main concern here".
>
> **A3**: The experiments in Appendix B.3.2, as what we further elaborated, have demonstrated how visually challenging the benchmark is. As shown in Fig. 12, **when images are replaced with their text descriptions**, the LVLMs' performances **stayed the same or improved for over 87%** of samples across all tested models (improved by ≥1 for **33-46%** samples and by ≥2 for **11-15%** samples). The observations can only be explained by **LVLMs struggle more with processing the raw visual input than with reasoning from a 'perfect' text description**. It leads to the evidence of the benchmark's visual challenge.

---

> ### Author Response · Authors · 2025-08-08
>
> ## About The Concerns in Visual Challenge (Part 2)
>
> **Q4**: "It is recommended to design a visual perception subtask, to ask the model the specific property that we want them to extract from the image, which can give people better understanding on model's main bottleneck of this benchmark."
>
> Thank you for the suggestion. Though MM-OPERA is an **open-ended** reasoning task where **any valid association is acceptable** (there is no predefined 'correct' detail the model must focus on), your suggestion inspired us to design a visual grounding diagnostic to better understand visual bottlenecks.
>
> We introduce a two-step **Visual Perception Subtask** in the Remote-Item Association (RIA) setting:
>
> 1. **Image Description**: The model is shown both images and asked to produce detailed, comprehensive descriptions.
>
> 2. **Text-only Reasoning**: The model receives only its own descriptions (not the images) and must complete the RIA task.
>
> We apply this to 400 randomly sampled items using GPT-4o and Gemini-2.5-Pro-Preview, comparing performance on the task using the original images versus using the model's own generated text descriptions. The A→B format compares performance: Original Task → Self-Gen-Text-Only Task. The final five columns (↑≥2 to ↓≥2) show the distribution of per-sample score changes—the percentage of samples that improved (↑), stayed the same (↔), or declined (↓) when images were described first.
>
> |Model|SR (%)|HR-4 (%)|HR-3 (%)|△HR (%)|↑≥2 (%)|↑1 (%)|↔0 (%)|↓1 (%)|↓≥2 (%)|
> |--|:--:|:--:|:--:|:--:|:--:|:--:|:--:|:--:|:--:|
> |GPT-4o|63.06→55.69|17.00→7.00|38.75→21.00|21.75→14.00|2.25|6.50|62.25|18.50|10.50|
> |Gemini-2.5-Pro-Preview|73.31→62.50|37.75→19.00|59.00→38.00|21.25→19.00|2.25|9.25|50.75|20.75|17.00|
>
> The results reveal that visual perception is a significant challenge. SR and HR scores drop notably when switching from images to self-generated descriptions. Crucially, the proportion of samples with a score decrease is **over three times higher** than those with an increase. This suggests that models **often miss critical visual details or introduce hallucinations during the description stage**, which harms downstream reasoning. The small percentage of improved cases is also expected: when visual descriptions are accurate, text-only associative reasoning can be easier than its multimodal counterpart.
>
> These findings help isolate failures in visual perception from those in reasoning. We will include them in the revision.
>
> ## About The Concerns in Motivation
>
> **Q5**: "such corrlation seems to be indirect since the examples in MM-OPERA are all real-world objects while scientific discovery often need the combination of new abstract concepts".
>
> **A5**: Beyond Reviewer's imagination, many scientfic discovery was achieved by linking an abstract concept and visual information. We would like to provide an example of DNA structure discovery:
>
> **Background:** By the 1950s, scientists knew that DNA was the carrier of genetic information, but its three-dimensional structure remained a profound mystery. Solving it was the key to understanding how heredity worked. Rosalind Franklin, a chemist and X-ray crystallographer, alongside her student Raymond Gosling, produced a high-quality X-ray diffraction image of a DNA crystal. This image, famously known as "Photo 51," displayed a distinct and clear 'X' shaped pattern.
>
> **Visual Association and Scientific Discovery:** For a trained crystallographer, this 'X' was not a random smudge; it was a powerful visual signal. When James Watson was shown this image, he immediately recognized its significance. This was a direct, knowledge-based **visual association**: the 'X' pattern is a classic signature produced when X-rays diffract off a **helical structure**. Watson and Francis Crick then used this core visual concept—the helix—as a scaffold for their thinking. They began to piece together a model, visually and spatially aligning the helical shape with other known chemical data. Their final model perfectly explained both the visual features of Photo 51 and all known chemical properties of DNA.
>
> **Q6**: Embodied-AI requires precise low-level visual perception and physics understanding, which is not included in the current benchmark.
>
> **A6**: Embodied-AI also requires high-level association instead of only low-level perception to achieve robotic intelligence rather than traditional robotic manipulation, e.g., how to use a unseen tool by associating its shape with functionality. To address the original concern by Reviewer rHto, we have provided the thorough discussion and possibility of future research derived from our benchmark. We are particularly confused why Reviewer rHto argued for Embodied AI because it is just the impact of our work instead of what we aim to do in this work.

---

### Note · Authors · 2025-08-14

This paper contributes the community on multimodal association reasoning, the foundamental ability that LVLM emerges yet existing benchmarks seldom investigated. Distinct from *Labyrinth of Links*, the first work that accessed this theme with **closed-ended** tasks, our work aims to benchmark the **open-ended** association reasoning (i.e., OEAR) in LVLM:

(1) It includes 11,497 task instances with the RIA and ICA designs inspired by psychometric tests, which were manually curated with reasoning paths, and span across 13 analytical dimensions with diverse fields, cultures, and languages (see Fig. 2).

(2) We propose process-rewarded LLM-as-a-Judge strategy that simultaneously evaluate each LVLM's response outcome and reasoning process for OEAR. It suits the open-ended setup with the impressive human-machine alignment and the resistance of LLM-as-a-Judge bias.

(3) The experimental analysis benchmarks LVLMs on OEAR, revealing significant gaps with humans and shedding key insights in creativity, conceptual distinctiveness, and robustness.

Our work was initially rated by 34455, where the work is praised to the novelty in addressing a key cognitive gap (pnjR, MHP7, 9E1P); the high quality and diversity (pnjR, i3RA, 9E1P); and the thoroughness of analysis (rHto, 9E1P).

We appreciate all invaluable feedbacks from all the reviewers and AC, and our rebuttals have addressed most concerns. While Reviewer rHto remains two concerns:

(1) rHto doubted **visual challenges in our benchmark** with *arbitrary and unjustified comments*, i.e., "sufficient to answer the question just with a few words", "I can not see any visual challenge presented". Instead, we tend to convince the reviewers and AC with evidences: first, difficulty$\neq$more words, and MM-OPERA has many cases with a short description yet hardly captured in vision; second, we present many instances that all LVLMs fail, and third, we reported the gap between text2text and image2text performances across 3 baselines in Fig. 13. Our evidences justify the visual challenges behind MM-OPERA.

(2) Reviewer rHto challenges the **application potential**, and we argue from two aspects. First, we have discussed 4 directions but rHto only challenges the cases in embodied AI and scentific discovery, but ignore the rest. Second, we have elaborated why OEAR work for embodied AI and scentific discovery with evidences, including the cases that rHto did not consider.

Regretfully, we did not receive his further response.

---

### Decision · Program_Chairs · 2025-09-18

**Decision:**

Accept (poster)

**Comment:**

The paper introduces MM-OPERA, a large-scale multimodal dataset designed to evaluate open-ended association reasoning in LVLMs, inspired by human cognitive tests like the Remote Associates Test. It features two tasks—Remote-Item Association (RIA) and In-Context Association (ICA)—that assess convergent and divergent thinking across 13 reasoning dimensions, 22 domains, and 15 languages, with 11,497 curated instances.

The main weaknesses raised by reviewers include concerns about the necessity of the visual modality, the real-world applicability of the task, and the scale of the human baseline. These concerns were effectively addressed in the rebuttal. The authors provided compelling new evidence—through a visual perception subtask and detailed case studies—that visual perception is a significant bottleneck, with performance dropping when models rely on generated descriptions instead of raw images. They clarified the motivation by linking associative reasoning to higher-order AI capabilities in scientific discovery, creative problem-solving, and education.


After carefully reading the original manuscript, the comments of each reviewer, and the author's response, I believe that the current manuscript meets the standards for acceptance because of (1) its significant contribution in creating a novel, large-scale benchmark for a critical yet underexplored capability—open-ended multimodal associative reasoning, (2) the rigorous and multi-faceted validation of its evaluation methodology, which demonstrates strong alignment with human judgment and mitigates common LLM-as-a-judge biases, and (3) the comprehensive experimental analysis that provides deep, diagnostic insights into the limitations of current LVLMs, revealing specific failure modes like perceptual misalignment and knowledge retrieval gaps. Thank you to all the reviewers for their thorough and constructive engagement, which has undoubtedly improved the quality of this work.